# Learning Koopman Representations with Controllability Guarantees

**Keyan Miao**[1*]    **Han Wang**[2*]    **Xuda Ding**[3]    **Konstantinos Gatsis**[4]
**Andreas Krause**[2]    **Antonis Papachristodoulou**[1]
[1]University of Oxford, UK    [2]ETH Zürich, Switzerland    [3]Bosch Central Research
[4]Villanova University, USA
{keyan.miao, antonis}@eng.ox.ac.uk    hanwang@control.ee.ethz.ch
konstantinos.gatsis@villanova.edu    krausea@ethz.ch

## Abstract

Learning nonlinear dynamical models from data is central to control. Two fundamental challenges exist: (1) how to learn accurate models from limited data, and (2) how to ensure the learned models are suitable for control design of the nominal system. We address both by enforcing a critical *a priori* property of the nominal system during learning: *controllability*. Controllability guarantees the existence of control policies that can drive the learned model from any initial state to any desired state. From a modeling perspective, it captures key structural features of the nominal system, thereby improving data efficiency. For downstream control, it enables the use of modern techniques such as model predictive control (MPC). Our approach is based on controllability-preserving Koopman representation learning. Rather than learning dynamics directly in the nominal state space, we learn in a latent space where the system admits a linear representation. We prove that controllability of the learned latent model implies controllability in the nominal state space. To enforce this property, we introduce a novel canonical parameterization of the latent dynamics matrices. We further incorporate Gramian-based regularization to shape the degree of controllability, yielding well-conditioned models for control. Implemented as an end-to-end Neural ODE framework, our method learns models that are both predictive and controllable from limited data. Experiments on nonlinear benchmarks demonstrate accurate long-horizon prediction, reliable MPC performance, and substantially improved data efficiency.

## 1 Introduction

Learning dynamical models from data is crucial for control design, analysis, and verification. For linear systems, methods such as ARX/ARMAX and subspace identification are well established, with strong theory and efficient algorithms (Ljung, 1998). Nonlinear extensions such as NARX (Billings, 2013), Volterra models, Hammerstein–Wiener structures (Billings, 1980), Gaussian processes (Kocijan, 2016), and grey-box approaches remain more challenging, as they often impose restrictive assumptions or scale poorly with system dimension.

Deep learning–based methods, including neural state-space models (Rangapuram et al., 2018), recurrent architectures (Hochreiter & Schmidhuber, 1997; Chung et al., 2014), and Neural ODEs (Chen et al., 2018; Rahman et al., 2022; Miao & Gatsis, 2023), offer expressive parameterizations capable of capturing complex nonlinear dynamics. While effective for trajectory prediction, the resulting models are often ill-suited for control: their nonlinear structure hinders the application of tools such as MPC, and they rarely provide guarantees on critical closed-loop properties such as safety and stability.

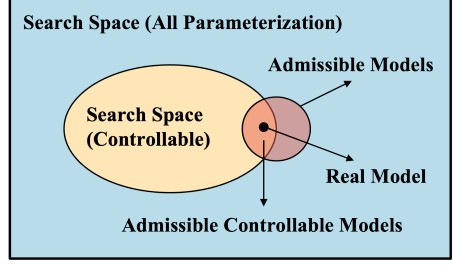

Figure 1: Encoding priors such as controllability reduces the search space

---

*Han Wang contributed equally to this work with Keyan Miao, serving as corresponding author.

A broader limitation of both classical and learning-based identification is the difficulty of incorporating *structural priors* of the nominal system. For control, one of the most critical priors is *controllability* (Klamka, 1963). As illustrated in Figure 1, restricting the search to controllable models greatly reduces the parameter space, thereby improving data efficiency. However, encoding controllability during training is challenging: even verifying it for nonlinear systems requires complex rank conditions over infinitely many Lie brackets (Isidori, 1985). Consequently, most learning procedures focus solely on trajectory fitting, leaving structural properties to be checked or enforced only after training. This disconnect often produces models that predict well but are unsuitable for control. While such analysis is routine for linear systems, it remains a major challenge in the nonlinear setting. A promising path to bridge this gap is through the Koopman operator.

Koopman-based modeling has long been explored in control theory to render nonlinear systems more amenable to analysis and feedback. The key idea is to approximate nonlinear dynamics with linear surrogates in a lifted space, enabling the use of linear systems theory (Koopman, 1931). Early methods such as dynamic mode decomposition (DMD) and its extensions relied on pre-specified basis functions (Williams et al., 2015; Kaiser et al., 2021; Brunton et al., 2022), while recent approaches integrate representation learning via autoencoders and neural networks. The common motivation is to obtain models that interface naturally with linear controllers, particularly MPC (Korda & Mezić, 2018). The appeal of Koopman representation learning lies in combining the expressiveness of nonlinear modeling with the tractability of linear control synthesis.

**Related work and Gap.** Learning-based Koopman approaches approximate nonlinear dynamics by training neural networks to construct observables, thereby lifting the system into a space where linear dynamics can be identified (Lusch et al., 2018; Yeung et al., 2019). The main benefit of lifting is that in the lifted space, traditional optimization based control methods can be easily implemented because of the linear dynamics (Korda & Mezić, 2018; Zinage & Bakolas, 2023). Existing variants differ in how operators are estimated (Han et al., 2020; Wang et al., 2021; Shi & Meng, 2022; Xiao et al., 2022) and in whether the operators are fixed or time varying (Li et al., 2025), yet the overall pipeline largely targets multi-step prediction. Structural properties, however, are seldom addressed. Controllability has been considered only rarely, typically by adding the Kalman rank condition of the lifted system as a loss (Han et al., 2020), which neither guarantees nor reflects the property. A recent paper (Choi et al., 2024) studied controllability preservation conditions, but it is limited to theoretical analysis under assumptions of exact representation. No computational methods were proposed in that work. Other priors have been explored even less. The first attempt incorporated *stabilizability* through LMI-based parameterizations (Fan et al., 2024), and a more recent direction embeds control inputs nonlinearly via neural networks (Guo et al., 2025), which requires post hoc Lie bracket checks and nonlinear optimization. In short, existing methods emphasize reconstruction but neither guarantee controllability nor exploit it as a structural prior to reduce the search space and guide training. This gap motivates our framework, which enforces controllability by construction, yielding models that are both predictive and reliable for control. Further discussion of related work is provided in Appendix A.

**Our contributions.** The contributions of this work are as follows:

- We propose a Koopman-based framework for learning nonlinear dynamical models within a neural ODE architecture. The approach yields linear surrogate models that are both accurate for prediction and efficient for control. The proposed framework accommodates irregular or multi rate sampling data and allows the learned continuous time dynamics to be used at control frequencies that differ from those in the training data without requiring modification or retraining.
- To ensure controllability of the learned model, we introduce a canonical parameterization of the Koopman operators. This guarantees controllability *by construction* in both single- and multi-input settings, while preserving expressiveness through learnable similarity transforms.
- To enhance control performance, we incorporate controllability Gramians into the training objective to increase the *degree of controllability*. By shaping their spectrum, we obtain better-conditioned models that reduce control effort in downstream tasks.
- We validate the framework on several nonlinear benchmarks. Experiments demonstrate improved data efficiency, higher prediction accuracy, and superior MPC performance, with greater feasibility and more reliable closed-loop behavior than unstructured baselines.

## 2 PRELIMINARIES

### 2.1 PROBLEM SETUP

We study the learning of nonlinear dynamical models. Specifically, we consider system

$$\dot{x}(t) = f(x(t), u(t)), \qquad x(t) \in \mathbb{R}^n, \ u(t) \in \mathbb{R}^m, \tag{1}$$

where $f(x, u)$ is unknown but locally controllable as *a priori*.

**Definition 1** (Controllability (Kalman et al., 1960)). *The control system* (1) *is said to have Controllability (or 'be controllable') if for any $x(t_0) = x_0 \in \mathbb{R}^n$ and $x_T \in \mathbb{R}^n$, there exists a continuous control signal $u(\cdot) : [t_0, t_f] \to \mathbb{R}^m$ such that*

$$x(t_f) = \int_0^{t_f} f(x(t), u(t))dt \bigg|_{x(t_0)=x_0} = x_T$$

The task is to construct, from data, a dynamical model $\hat{f}(x(t), u(t))$ that accurately captures the input–state evolution (1) and remains *controllable* for control design. This entails not only performing supervised learning but also a mechanism that preserves controllability in the learned model $\hat{f}(x(t), u(t))$.

Data of (1) is sampled as a time series from the initial state $x_0$, using a continuous control signal $u(\cdot) : [0, d_K] \to \mathbb{R}^m$:

$$\mathcal{X} := [x(0) \quad x(d_1) \quad x(d_2) \quad \dots \quad x(d_K)]$$
$$\mathcal{U} := [u(0) \quad u(d_1) \quad u(d_2) \quad \dots \quad u(d_K)]$$

### 2.2 NEURAL ODES WITH INPUT

Neural Ordinary Differential Equations (ODEs) (Chen et al., 2018) extends the idea of deep residual networks (He et al., 2016). It has shown great success in learning continuous time dynamical models.

**Definition 2** (Neural ODE with input). *With $h_x : \mathbb{R}^{n_x} \to \mathbb{R}^{n_z}$, $h_y : \mathbb{R}^{n_z} \to \mathbb{R}^{n_y}$ representing the input network and output network respectively, a Neural ODE with input is a system of the form*

$$\begin{cases} z(t_0) = h_x(x(t_0)) \\ \dot{z}(t) = \mathcal{F}(t, z(t), u(t), \theta), \quad t \in \mathcal{S} \\ y(t) = h_y(z(t)) \end{cases} \tag{2}$$

*where $\mathcal{S} := [t_0, t_f]$ $(t_0, t_f \in \mathbb{R}^+)$ is the depth domain and $\mathcal{F}$ is a neural network referred to as ODENet with parameter $\theta$; $u(t)$ is the input at time $t$.*

The terminal state $x(t_f)$, obtained by solving the initial value problem (IVP), represents the evolved system state. In Neural ODEs, depth corresponds to continuous evolution over time, with ResNets interpretable as Euler discretizations. To encode structural priors into the learned model, we adopt the Koopman representation framework, which introduces a *linear map* $\mathcal{F}(\cdot)$.

### 2.3 KOOPMAN OPERATOR

For the nonlinear control system (1), the goal of Koopman-based modeling is to obtain a linear representation of the dynamics in a higher-dimensional space of observables. In this setting, the nonlinear evolution of $(x, u)$ is described by a linear operator acting on functions of $(x, u)$. Koopman operator theory establishes the existence of an infinite-dimensional operator $\mathcal{K}$ acting on observables $\Psi(x, u)$:

$$\frac{d}{dt} \Psi(x(t), u(t)) = \mathcal{K} \Psi(x(t), u(t)).$$

In applications, it usually uses a finite-dimensional approximation for the Koopman operator on a finite-dimensional function space. A standard construction is to select observables of the form:

$$\Psi(x, u) = \begin{bmatrix} z \\ u \end{bmatrix}, \quad z = \phi(x) \in \mathbb{R}^N,$$

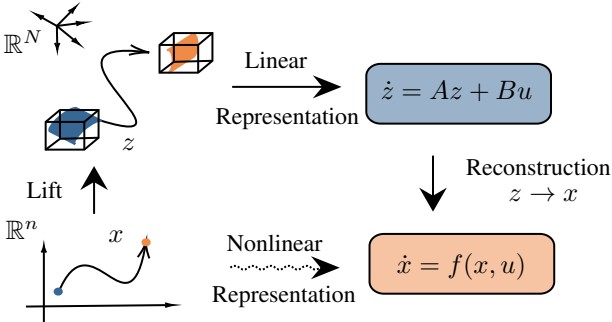

Figure 2: Koopman-based dynamical model representation.

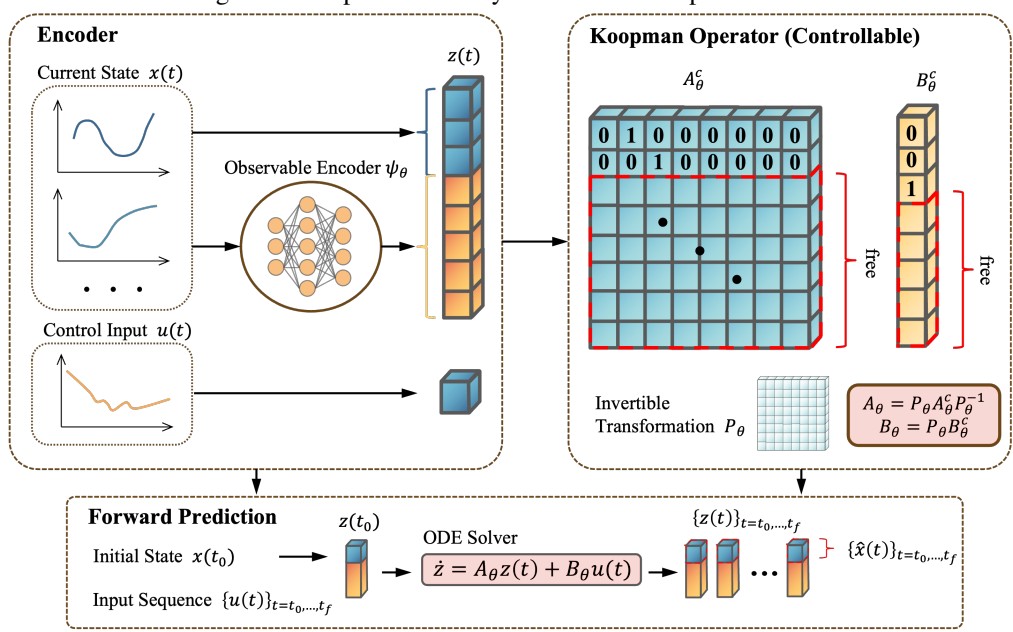

Figure 3: Illustration of the Koopman based Neural ODE learning pipeline.

where $\phi : \mathbb{R}^n \to \mathbb{R}^N$ with $N \gg n$ is a set of nonlinear lifiting functions applied to the state $x$, and the input $u$ is retained in its original coordinates. Since prediction and control only require the lifted state dynamics, it suffices to retain the first $N$ rows of this operator. This leads to the following Koopman representation as an ODE:

$$\dot{z}(t) = Az(t) + Bu(t), \quad \hat{x}(t) = h(z(t)), \quad z(t) = \phi(x(t))$$

In many applications, $h$ is chosen to be linear, i.e., $h(z) = Cz$ with $C \in \mathbb{R}^{n \times N}$. The functionality of the Koopman operator is illustrated in Figure 2, with further details provided in Appendix B.1.

## 3 KOOPMAN LEARNING FRAMEWORK

Building on the integration of the Koopman operator with Neural ODEs, we propose a Koopman-based system learning framework, illustrated in Figure 3. This end-to-end pipeline couples a neural encoder, a controllable Koopman operator, and a differentiable ODE solver. The goal is to learn a Neural ODE of the form

$$\dot{z}(t) = A_\theta z(t) + B_\theta u(t), \quad z(t) = \phi_\theta(x(t)) = [x(t)\ \psi_\theta(x(t))], \quad \hat{x}(t) = y(t) = Cz(t), \quad (3)$$

that guarantees controllability while modeling the unknown but controllable dynamics $\dot{x} = f(x, u)$.

The input network $h_x$ is specified as the observable encoder $\phi_\theta$ in the "Encoder" block. To preserve the nominal state explicitly, the first $n$ components of $\phi_\theta$ are included through identity lifting

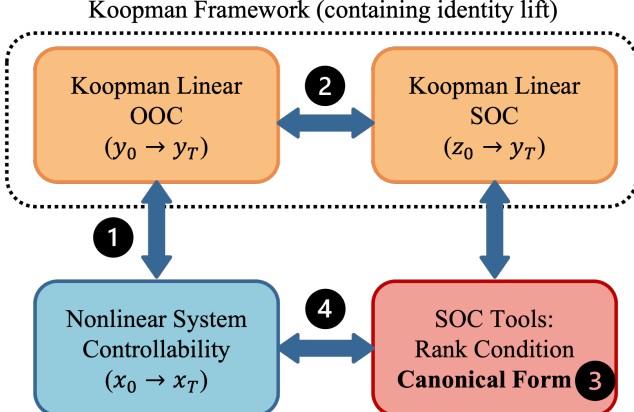

Figure 4: Overview of the relations among controllability notions. ① We show that the controllability of the original nonlinear system is equivalent to OOC of Koopman linear representation (in main text); ② In the Koopman linear representation, we prove that OOC coincides with SOC (Lemma 1and Theorem 1, proved in Appendix D); ③ Based on this equivalence, we for the first time introduce a SOC canonical form which can be used for OOC canonical form for Koopman representation (Theorem 2, proved in Appendix D); ④ Finally, we leverage this canonical form to parameterize the neural Koopman operator, which enables constructing learning-based Koopman representations of the original nonlinear system.

$(x(t) \in \mathbb{R}^n)$, while the remaining $N - n$ entries are nonlinear observables $\psi_\theta(x)$ to be learned. The "Koopman Operator (Controllable)" block specifies the structure of $A_\theta$ and $B_\theta$ that enforces controllability of the learned Neural ODE. The "Forward Prediction" block follows standard practice: given an initial state $x_0$ and input signal, the ODE solver propagates the dynamics to obtain the terminal state.

Finally, to recover the nominal state $x(t)$, we define the output network as $h_y(z) = Cz$ with

$$\hat{x}(t) = y(t) = Cz(t), \quad C = [I_n \quad 0],$$

so that $x(t)$ is obtained directly as the output of the Neural ODE.

## 3.1 CONTROLLABILITY ANALYSIS

Given that the nominal nonlinear system (1) is controllable and the learned Neural ODE (3) yields the output $\hat{x} = y = Cz$, we ask whether the system can be driven from any initial condition to any desired target output $y(t_f)$. This consideration motivates the following definition.

**Definition 3** (State-to-Output Controllability (SOC)). *System (3) is said to be* state-to-output controllable *on* $(\mathcal{Z} \subseteq \mathbb{R}^N, \mathcal{Y} \subseteq \mathbb{R}^n)$ *if, for any initial state condition* $z(t_0) = z_0 \in \mathcal{Z}$ *and a target output* $y_T \in \mathcal{Y}$, *there exists a continuous input signal* $u : [t_0, t_f] \to \mathbb{R}^m$ *such that* $y(t_f) = y_T$.

**Definition 4** (Output-to-Output Controllability (OOC)). *System (3) is said* output-to-output controllable *on* $\mathcal{Y} \subseteq \mathbb{R}^n$ *if, for any initial output* $y(t_0) = y_0 \in \mathcal{Y}$ *and target output* $y_T \in \mathcal{Y}$, *there exists a continuous input signal* $u : [t_0, t_f] \to \mathbb{R}^m$ *such that* $y(t_f) = y_T$.

We are interested in OOC and SOC for system (3) because its output $y$ recovers the state $x$ of the original nonlinear system. If (3) is OOC, then the corresponding nonlinear system is *controllable* in the sense of Definition 1. However, verifying OOC is generally more difficult than SOC, which can be checked using the Kalman rank condition, no comparably simple algebraic test exists for OOC (Danhane et al., 2023). Within our Koopman representation framework with identity lift, we show that SOC and OOC coincide in the following lemma.

**Lemma 1.** *Consider system (3). Define the set* $\mathcal{Z} \subset \mathbb{R}^N$ *to be such that* $\mathcal{Z} := \phi(\mathbb{R}^n)$. *Then, the system is OOC on* $\mathbb{R}^n$ *if and only if it is SOC on* $(\mathcal{Z}, \mathbb{R}^n)$.

While verifying OOC is generally difficult for linear systems, Lemma 1 establishes the equivalence between OOC on $\mathbb{R}^n$ and SOC on $(\mathcal{Z}, \mathbb{R}^n)$. This equivalence allows us to verify OOC of system (3)

through SOC. OOC then implies controllability as per Definition 1 for the corresponding learned nonlinear system. The verification criterion is stated in the following theorem.

**Theorem 1.** *The system* (3) *is OOC on* $\mathbb{R}^n$ *if and only if the controllability matrix*

$$\mathcal{C} = C[B_\theta \quad A_\theta B_\theta \quad \ldots \quad A_\theta^{N-1}B_\theta] \tag{4}$$

*is full-rank.*

The proofs are provided in Appendix D. From a training perspective, one could promote SOC of the Koopman linear system by adding a loss term of the form $-\min \operatorname{eig}(\mathcal{C})$. However, as in many learning problems, this approach provides no guarantees. To address these issues, we propose a *direct parameterization* strategy that explicitly constrains the matrices $A_\theta$ and $B_\theta$, ensuring that the learned Neural ODE (3) is always OOC. An overview of the conceptual framework and our contributions is provided in Figure 4 in Appendix C.

## 3.2 CONTROLLABILITY CANONICAL FORM

The following theorem shows the parameterization for $A_\theta$ and $B_\theta$ that ensures OOC of (3).

**Theorem 2.** *Consider system* (3). *Then, the system is OOC on* $\mathbb{R}^n$ *if there exist matrices* $A_\theta^c \in \mathbb{R}^{N \times N}$, $B_\theta^c \in \mathbb{R}^{N \times m}$ *and* $P_\theta \in \mathbb{R}^{N \times N}$, *where*

$$A_\theta^c = \begin{bmatrix} 0 & 1 & 0 & \ldots & \ldots & 0 \\ 0 & 0 & 1 & 0 & \ldots & 0 \\ \vdots & \vdots & \ddots & \ddots & \ddots & \vdots \\ 0 & 0 & \ldots & 1 & \ldots & 0 \\ a_1 & a_2 & \ldots & \ldots & \ldots & a_N \\ b_1 & b_2 & \ldots & \ldots & \ldots & b_N \\ \vdots & \vdots & \ddots & \ddots & \ddots & \vdots \\ c_1 & c_2 & \ldots & \ldots & \ldots & c_N \end{bmatrix} \quad B_\theta^c = \begin{bmatrix} 0 \\ 0 \\ \vdots \\ 0 \\ 1 \\ d_1 \\ \vdots \\ d_{N-n} \end{bmatrix} \quad P_\theta = \operatorname{diag}(P_1, P_2), \tag{5}$$

*such that*

$$A_\theta = P_\theta A_\theta^c P_\theta^{-1} \quad B_\theta = P_\theta B_\theta^c \tag{6}$$

$P_1 \in \mathbb{R}^n$ *and* $P_2 \in \mathbb{R}^{N-n}$ *are both full rank matrices. The first* $n-1$ *rows of* $A_\theta$ *only consists of* 0 *and* 1, *and all* 1s *are on the superdiagonal. All the other elements of* $A_\theta$ *are free. The first* $n-1$ *rows of* $B_\theta$ *is* 0, *the* $n$th *row is* 1, *and all the other rows are free.*

The proof is provided in Appendix D. Equipped with Theorems 1 and 2, we can learn a Neural ODE (3) that models the original nonlinear system (1) while preserving controllability. It is important to note that Theorem 2 applies to single-input systems ($m = 1$). The results can be extended to multi-input settings via the Brunovský decomposition, as discussed in Appendix E.1.

## 3.3 DEGREE OF CONTROLLABILITY VIA GRAMIANS

Beyond the binary notion of controllability, the degree of controllability determines the energy required to steer the system (Kailath, 1980). To quantify this, we use the finite-horizon output Gramian

$$W_T^y = \int_0^T (Ce^{A_\theta \tau}B_\theta)(Ce^{A_\theta \tau}B_\theta)^\top \, d\tau,$$

which measures how inputs excite the physical state over a finite time window. A small $\lambda_{\min}(W_T^y)$ indicates directions in the state space that require disproportionately high input energy to reach. A large condition number $\kappa(W_T^y)$ implies unbalanced controllability and ill-conditioned optimization problems. To address this, we introduce the regularizer

$$\mathcal{R}_{\mathrm{gram}}(A_\theta, B_\theta) = \frac{1}{\lambda_{\min}(W_T^y)} + \gamma \kappa(W_T^y),$$

with $\gamma > 0$ a trade-off parameter. This term encourages a larger smallest eigenvalue while discouraging large condition numbers, promoting balanced controllability across directions.

---

**Algorithm 1:** Training Neural ODEs based Koopman representation with controllability guarantees

---

**Input:** Dataset $\mathcal{D}$, horizon $[t_0, t_f]$, decay $\eta$, Gramian horizon $T$, weight $\lambda_{\text{gram}}$
**Output:** Parameters $\theta$
Initialize $\theta = \{\theta_\phi, \theta_{A_\theta^c}, \theta_{B_\theta^c}, \theta_P\}$;
**for** *epoch = 1 to E* **do**
    Sample mini-batch $\mathcal{B} \subset \mathcal{D}$;
    **for** *trajectory $(x, u) \in \mathcal{B}$* **do**
        $z(t_0) \leftarrow \Psi_\theta(x(t_0))$ ;                        `// Encode initial state`
        $(A_\theta, B_\theta) \leftarrow P_\theta^{-1}(A_\theta^c, B_\theta^c)P_\theta$ ;        `// Koopman dynamics`
        $z(t) \leftarrow \text{ODESolver}(\dot{z} = A_\theta z + B_\theta u), [t_0, t_f])$ ; `// Rollout in lifted space`
        $\hat{x}(t) \leftarrow Cz(t)$ ;                        `// Recover state prediction`
        $\mathcal{L}_{\text{pred}} \leftarrow \int w(t)\|\hat{x}(t) - x(t)\|^2 dt$ ;        `// Trajectory loss`
        $\mathcal{R}_{\text{gram}} \leftarrow \text{Regularizer}(W_T^y)$ ;            `// Gramian shaping`
        $\mathcal{L} \leftarrow \mathcal{L}_{\text{pred}} + \lambda_{\text{gram}}\mathcal{R}_{\text{gram}}$ ;            `// Total loss`
    $\theta \leftarrow \text{Optimizer}(\theta, \nabla_\theta \mathcal{L})$ ;               `// Parameter update`

---

### 3.4 LOSS FUNCTIONS DESIGN

The state matrices $A_\theta$ and $B_\theta$ are parameterized as in (6), where $(A_\theta^c, B_\theta^c)$ denotes the canonical controllable form from (5), and $P_\theta$ is a trainable invertible transformation. This construction preserves controllability by design while maintaining flexibility: without $P_\theta$, the input would have only limited influence on the identity coordinates.

Given an input trajectory $u : [t_0, t_f] \to \mathbb{R}^m$ and initial condition $x(t_0)$, the Neural ODE dynamics (3) with initial state $z(t_0) = \Psi_\theta(x(t_0))$ generate a predicted state trajectory $\hat{x}(t) = Cz(t)$. We define the prediction loss as

$$\mathcal{L}_{\text{pred}}(\theta) = \frac{1}{t_f - t_0} \int_{t_0}^{t_f} w(t) \|\hat{x}(t) - x(t)\|_2^2 \, dt,$$

where $w(t)$ is a nonnegative weight function. This loss evaluates accuracy over the entire rollout rather than only at the next step, thereby encouraging stable long-horizon predictions. We choose $w(t)$ with a decay rate that emphasizes early prediction errors while still accounting for the full trajectory. The integral is computed in discretized form, with the discretization determined by the dataset's sampling scheme.

To complement predictive accuracy with controllability, we augment this loss with the Gramian regularizer from Section 3.3, yielding the full training objective

$$\min_\theta \quad \mathcal{L}_{\text{pred}}(\theta) \; + \; \lambda_{\text{gram}} \, \mathcal{R}_{\text{gram}}(A_\theta, B_\theta).$$

Gradients are computed through the ODE solver, enabling an end-to-end formulation that yields models accurate over long horizons, controllable by design, and numerically well conditioned for downstream control.

### 3.5 MODEL PREDICTIVE CONTROL ON LEARNED MODELS

We deploy MPC on the learned Neural ODE (3) by discretizing it with sampling time $\Delta t$ under zero-order hold:

$$A_\theta^d = e^{A_\theta \Delta t}, \qquad B_\theta^d = \int_0^{\Delta t} e^{A_\theta \tau} B_\theta \, d\tau, \qquad x_k = Cz_k.$$

At time $k$ we solve the following finite-horizon MPC on the discrete-time surrogate:

$$\min_{\{u_{j|k}\}_{j=k}^{k+H-1}} \sum_{j=k}^{k+H-1} \left( \|Cz_{j|k} - x^{\mathrm{ref}}\|_Q^2 + \|u_{j|k} - u_{j-1|k}\|_R^2 \right) + \|Cz_{k+H|k} - x^{\mathrm{ref}}\|_P^2, \quad (7)$$

$$\text{s.t. } z_{j+1|k} = A_\theta^d z_{j|k} + B_\theta^d u_{j|k}, \quad j = k, \ldots, k+H-1, \tag{7a}$$

$$z_{k|k} = \Psi_\theta(x_k), \tag{7b}$$

$$Cz_{j|k} \in \mathcal{X}, \quad u_{j|k} \in \mathcal{U}. \tag{7c}$$

Here $x^{\mathrm{ref}}$ is a reference in the physical coordinates, and $Q, R, P \succeq 0$ are standard MPC weights; the input-smoothing term $\|u_{j|k} - u_{j-1|k}\|_R^2$ can be replaced by $\|u_{j|k}\|_R^2$ if desired. Since the optimization is posed entirely on the linear surrogate $(A_\theta^d, B_\theta^d, C)$, (7) is a convex quadratic program. At each sampling instant we apply the first control $u_{k|k}^\star$, measure $x_{k+1}$, reset $z_{k+1|k+1} = \Psi_\theta(x_{k+1})$, and repeat in the receding-horizon fashion.

## 4 EMPIRICAL EVALUATION

The experimental evaluation is designed to assess both model learning accuracy and control performance. We first study widely used nonlinear control benchmarks: pendulum swing-up, mountain car, and cartpole stabilization.

To evaluate data efficiency, we train each model with different fractions of the available dataset $(1\%, 5\%, 10\%, 30\%, 50\%, 100\%)$ and measure both short- and long-horizon prediction errors. The first baseline is the Deep Koopman Operator (DKO) (Han et al., 2020; Wang et al., 2021; Xiao et al., 2022; Shi & Meng, 2022), which employs deep neural networks to learn Koopman representations. Although existing variants differ in technical details, their overall pipeline is similar; in our experiments we adopt a continuous-time realization. The second baseline is a multilayer perceptron (MLP) (Chua et al., 2018) trained directly on the nonlinear dynamics, representing a purely data-driven approach without structural priors.

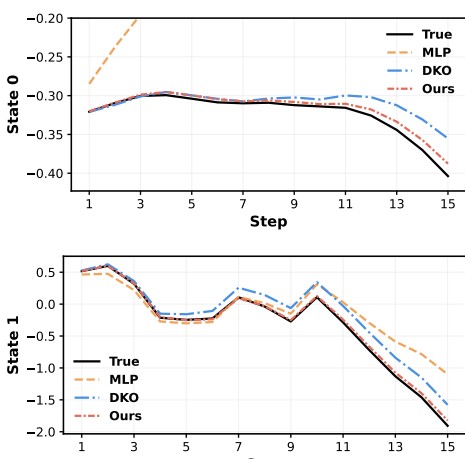

Figure 5: 15 Steps Prediction Results on test case of Pendulum

We compare these against our method on (i) trajectory prediction error, (ii) cumulative input energy for control. These metrics reflect both the model fitting ability and the downstream control utility of the learned dynamics. In the end, we extend our framework to the multi-input setting and validate it on a six-dimensional gene regulatory network (GRN) (Elowitz & Leibler, 2000) system. All experimental settings, including dataset generation, training details, and hyperparameters, are described in Appendix G.

**Prediction performance**  Table 1 reports model learning errors across varying training dataset sizes. Our framework consistently outperforms both DKO and MLP when less than $30\%$ of the data is available. With abundant data, our framework and DKO achieve comparable accuracy, indicating that both are sufficiently expressive to capture the underlying dynamics. The key distinction is that our method guarantees controllability by construction, whereas DKO can only assess it *a posteriori* without guarantees during training. Figure 5, trained with only $30\%$ of the pendulum dataset, shows that our model closely tracks the true trajectory, while the baselines deviate despite using the same data. This advantage stems from enforcing controllability as a structural prior, which reduces the parameter search space and guides optimization toward meaningful solutions. Consequently, our model attains lower errors with fewer samples and converges faster during training. The training curves in Figure 6 support this view. Our losses decrease rapidly and stabilize at a lower level, whereas DKO starts with large errors and converges slowly. Although nonlinear, the discrete MLP

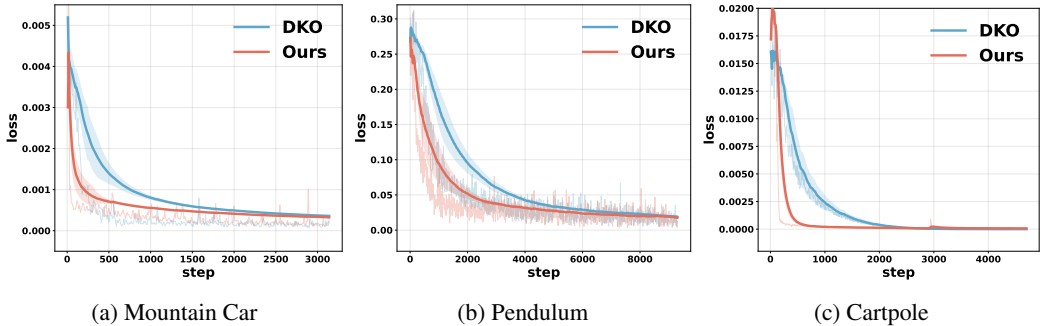

(a) Mountain Car      (b) Pendulum      (c) Cartpole

Figure 6: Loss versus training step on Mountain Car, Pendulum, and Cartpole. Our method converges faster than DKO.

Table 1: Comparison of prediction error (MSE) across environments with varying fractions of training data. Our method shows clear advantages under limited training data, highlighting its data efficiency.

| Environment | Method | 1% | 5% | 10% | 30% | 50% | 100% |
|---|---|---|---|---|---|---|---|
| Mountain Car | MLP | 0.0758 | 0.0571 | 0.0178 | 0.00510 | 0.00418 | 0.0002 |
| | DKO | 0.0200 | 0.00022 | 0.00016 | | $\leq 1 \times 10^{-4}$ | |
| | Ours | **0.0032** | **0.00019** | **0.00011** | | | |
| Pendulum | MLP | 1.1778 | 0.5998 | 0.2610 | 0.0470 | 0.0218 | 0.0091 |
| | DKO | 1.5347 | 0.1079 | 0.0390 | 0.0086 | $\leq 6 \times 10^{-3}$ | |
| | Ours | **0.3747** | **0.0318** | **0.0114** | **0.0061** | | |
| Cartpole | MLP | 0.0465 | 0.0296 | 0.0202 | 0.00727 | 0.0039 | 0.0021 |
| | DKO | 0.1306 | 0.01452 | 0.007585 | 0.00064 | $\leq 5 \times 10^{-4}$ | |
| | Ours | **0.0095** | **0.0024** | **0.001153** | **0.000571** | | |

baseline struggles to capture continuous dynamics, leading to rapid error accumulation over rollouts. In summary, controllability provides not only a downstream guarantee but also a powerful inductive bias that improves data efficiency, accelerates convergence, and stabilizes prediction.

**Control Performance** We next evaluate control performance by embedding the learned models into MPC. As shown in Table 2, differences in prediction accuracy translate directly into control outcomes. With limited training data, DKO often fails to complete the task due to inaccurate or uncontrollable models that render MPC infeasible. In contrast, by encoding controllability as a structural prior, our method learns more accurate models from scarce data and produces MPC solutions that require substantially less input energy.

Figure 7 illustrates this effect in the mountain car task. With only 1% of the dataset, DKO fails to stabilize the car under MPC, whereas our model successfully drives the system to the target. Even with 5% of the data, our approach requires substantially less control effort than DKO, highlighting the advantages of accurate and controllable dynamics. Similar trends are observed in pendulum swing-up and cartpole stabilization.

Another line of work introduces time-varying Koopman operators (Li et al., 2025), where the state matrix $A$ is generated at each step from past trajectories rather than fixed. This increases flexibility and can improve prediction, but at the cost of higher computational burden for MPC, since $A$ must be recomputed at each step. Moreover, since trajectory history is required (e.g., 30 prior steps), a real system must first be driven with preliminary inputs before control actions can be computed, complicating deployment. Controllability analysis is also more challenging for time-varying Koopman models. General nonlinear models such as Neural ODEs and LSTMs can achieve strong predictive accuracy by directly parameterizing the dynamics. Their limitation lies in downstream use: embedding them in MPC typically leads to nonconvex optimization problems that are much slower to solve and lack convergence guarantees, unlike linear formulations.

Table 2: MPC input cost comparison at selected dataset fractions. "Fail" indicates that the controller was unable to complete the task.

| Environment | Task | Data | DKO Cost | Our Cost | Rel.↓ (%) |
|---|---|---|---|---|---|
| Mountain Car | Hilltop | 1% | Fail | **186.59** | – |
|  |  | 5% | 297.51 | **165.21** | 44.5 |
| Pendulum | Swing-up & hold | 5% | Fail | **100.25** | – |
|  |  | 10% | 239.2 | **43.50** | 81.8 |
| Cartpole | Balance | 30% | Fail | **19.61** | – |

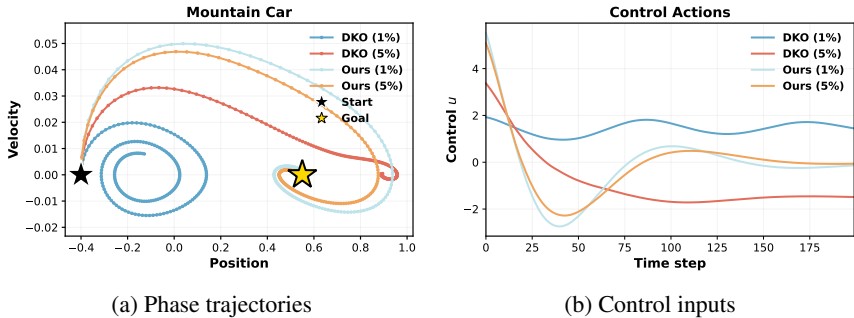

(a) Phase trajectories          (b) Control inputs

Figure 7: Comparison of Mountain Car experiments on control task.

**Multi-input extension.** We extend our framework to the multi-input setting and validate it on other more complex robotics environments including Reacher (Mujoco) and 7-DoF Franka manipulator environment used in Shi & Meng (2022). As shown in Appendix G, the results demonstrate that our method can scale to larger and multi-input systems and still show higher data efficiency. We also extend our approach on a six-dimensional gene regulatory network (GRN) (Elowitz & Leibler, 2000) with three control inputs. Under the same protocol as the single-input tasks, our controllability-preserving Koopman model attains a test MSE of $1.0 \times 10^{-4}$, compared to $3.0 \times 10^{-4}$ for DKO. On the associated control task, where one state must be regulated at a setpoint of 6, our method achieves an input cost of 80.96, lower than the 89.98 required by DKO. These results demonstrate that the proposed canonical parameterization and training procedure scale beyond single-input systems, while also highlighting the added challenges of multi-input settings, where optimization is harder to stabilize and performance degrades more quickly with limited data.

## 5   CONCLUSION

We proposed an end-to-end framework for learning nonlinear dynamical models with controllability guarantees, built on Koopman representations learned through Neural ODEs. To ensure controllability, we introduced novel OOC controllability canonical forms that parameterize the state matrices of the Koopman surrogate, providing guarantees *by construction*. Simulations on nonlinear benchmarks demonstrate superior data efficiency and control performance compared to state-of-the-art baselines. At the same time, there are some open challenges: multi-input controllability introduces additional complexity in initialization and parameterization, and in practice it is often unclear how to assign control inputs to state variables. Addressing these issues is an important direction for future work, and progress here could significantly broaden the applicability of controllability-aware system identification.

### ACKNOWLEDGMENTS

This work was supported by NCCR Automation, grant agreement 51NF40_225155 from the Swiss National Science Foundation. This support funded Keyan Miao's research visit at the ETH AI Center, during which part of this work was carried out. Keyan Miao was also supported by the Engineering and Physical Sciences Research Council (EPSRC) through grant EP/T517811/1, and

Antonis Papachristodoulou through the EEBio Programme Grant EP/Y014073/1, EP/X031470/1 and UKRI2108. The authors would also like to thank Dr. Armin Lederer for helpful discussions and insightful feedback.

**Reproducibility Statement.** Source code can be found at https://github.com/KYMiao/Controllable-Koopman; for theoretical results, clear explanations of any assumptions and a complete proof of the claims can be found in the Appendix D; for any datasets used in the experiments, a complete description of the data processing steps can be found in Appendix G.

**Ethics Statement** This work involves only simulated environments and raises no ethical concerns.

**Use of LLMs** We used a large language model (LLM) solely to aid and polish the writing and improve the clarity of exposition. All research ideas, technical contributions, experiments, and analyses are our own.

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

# A  DETAILED RELATED WORK

Learning-based Koopman approaches (Lusch et al., 2018; Yeung et al., 2019; Ricardo Constante-Amores et al., 2024) approximate nonlinear dynamics by training neural networks to construct observables, thereby lifting the system into a space where linear dynamics are identified. Despite differences in implementation, the pipeline is largely consistent: an encoder maps the state into a latent space, and operators $(A, B)$ are then estimated either through direct regression (e.g., least squares) or as trainable parameters within an end-to-end framework (Han et al., 2020; Wang et al., 2021; Shi & Meng, 2022; Xiao et al., 2022). Variants further differ in whether the operators are kept fixed or allowed to vary with time (Li et al., 2025), but the overall objective remains multi-step trajectory reconstruction. Crucially, structural properties of the underlying system are rarely incorporated.

Very few works have even attempted to consider controllability in this setting, and those that do typically include the Kalman rank condition as a loss term (Han et al., 2020). This suffers from two major issues. First, placing the condition in the loss provides no guarantee of preserving controllability. Second, and more importantly, requiring controllability of the entire lifted system is unrealistic and misguided: the lift is often very high dimensional while the input dimension remains small, making global controllability virtually impossible; moreover, in control tasks the relevant quantity is the reachability of the original physical state, not arbitrary auxiliary observables. As a result, such lifted-space conditions fail to capture the property that actually matters.

Research on incorporating structural priors into Koopman learning remains limited. The first such attempt considered *stabilizability*, introducing LMI-based parameterizations of the Koopman operator (Fan et al., 2024). This was the earliest demonstration that parameterization can embed system-theoretic properties into Koopman models, but stabilizability is a weaker requirement than controllability and cannot guarantee capabilities such as trajectory tracking. It is also worth noting that, more broadly, some recent works have introduced properties such as stability, stabilizability, or dissipativity into dynamics or control learning, for example via Neural System Level Synthesis (Furieri et al., 2022), or SDP/QP formulations (Yin et al., 2021; Tang et al., 2024; Okamoto & Kojima, 2025). While these demonstrate that inductive biases can be beneficial, such approaches either impose heavy computational burdens through projection-based optimization, or rely on the availability of base stabilizing controllers, and in any case highlight that structural priors are gaining attention while simple and efficient mechanisms remain highly needed.

A very recent effort, parametric Koopman models, embeds the control input nonlinearly via neural networks (Guo et al., 2025). Here controllability can only be verified post hoc using Lie bracket conditions, and because the input enters nonlinearly, the resulting control problems remain nonlinear optimizations, limiting the practical benefit of the linear surrogate.

In summary, existing Koopman learning methods focus primarily on reconstruction accuracy, with occasional efforts to enforce stabilizability or to post hoc analyze controllability. None provide guarantees of controllability during training, nor do they shape its degree in a way that affects the feasibility and efficiency of MPC. Moreover, by neglecting controllability as a structural prior, they miss the opportunity to reduce the effective parameter search space and to guide the training process itself. This gap motivates our framework, which enforces controllability by construction and explicitly regularizes the degree of reachability, yielding models that are both predictive and reliable for control.

## B   DETAILED PRELIMINARIES

### B.1   KOOPMAN OPERATOR

To handle inputs explicitly, the system can be reformulated on an augmented state space,

$$\mathcal{X}(t) = \begin{bmatrix} x(t) \\ u(t) \end{bmatrix}, \qquad \frac{d}{dt}\mathcal{X}(t) = \mathcal{F}(\mathcal{X}(t)) = \begin{bmatrix} f(x(t), u(t)) \\ \dot{u}(t) \end{bmatrix}.$$

The dynamics are autonomous in $\mathcal{X}(t)$, so one can define a Koopman operator acting on observables $\Psi : \mathbb{R}^{n+m} \to \mathbb{R}^{\tilde{N}}$. This operator is linear but infinite-dimensional, and satisfies

$$\frac{d}{dt}\Psi(\mathcal{X}(t)) = \mathcal{K}\Psi(\mathcal{X}(t)).$$

In other words, the nonlinear flow of $(x, u)$ induces linear dynamics in the lifted coordinates $\Psi(\mathcal{X})$.

Since $\mathcal{K}$ is infinite-dimensional, a finite-dimensional approximation is required for practical use. A standard construction is to select observables of the form

$$\Psi(\mathcal{X}) = \begin{bmatrix} \phi(x) \\ u \end{bmatrix}, \qquad z = \phi(x) \in \mathbb{R}^N,$$

where $\phi : \mathbb{R}^n \to \mathbb{R}^N (N \gg n)$ is a set of nonlinear lifting functions applied to the state, and the input $u$ is retained in its original coordinates. With this choice, the operator reduces to a block-matrix structure,

$$\frac{d}{dt}\begin{bmatrix} z(t) \\ u(t) \end{bmatrix} = K_\Psi \begin{bmatrix} z(t) \\ u(t) \end{bmatrix}, \qquad K_\Psi = \begin{bmatrix} A & B \\ * & * \end{bmatrix}.$$

Here $A \in \mathbb{R}^{N \times N}$ governs the autonomous dynamics of the lifted state, $B \in \mathbb{R}^{N \times m}$ describes how control inputs affect the lifted state, and the lower blocks (denoted by $*$) capture the evolution of the input coordinates themselves. Since prediction and control only require the lifted state dynamics, it suffices to retain the first $N$ rows of this operator. This leads to the following Koopman representation as an ODE:

$$\dot{z}(t) = Az(t) + Bu(t), \quad \hat{x}(t) = h(z(t)), \quad z(t) = \phi(x(t))$$

The tuple $(\phi, A, B, h)$ constitutes a finite-dimensional approximation of the infinite-dimensional controlled Koopman operator. In many applications $h$ is taken to be linear, i.e. $h(z) = Cz$ with $C \in \mathbb{R}^{n \times N}$.

### B.2   CONTROLLABILITY OF LINEAR SYSTEMS

Consider the linear control system defined globally

$$\dot{z}(t) = Az(t) + Bu(t), \qquad z(t) \in \mathbb{R}^N, \ u(t) \in \mathbb{R}^m,$$

with $A \in \mathbb{R}^{N \times N}$ and $B \in \mathbb{R}^{N \times m}$.

Dislike the challenging nonlinear controllability verification problem, the problem is tractable for linear systems.

**Proposition 1** (Kalman Rank Condition). *The linear system with matrices $(A, B)$ is controllable if and only if the controllability matrix*

$$\mathcal{C} = \begin{bmatrix} B & AB & A^2B & \cdots & A^{N-1}B \end{bmatrix}$$

*has full row rank, i.e.* $\mathrm{rank}(\mathcal{C}) = N$. *In this case, the columns of $\mathcal{C}$ span the entire state space, so any state can be reached from any initial condition by a suitable input.*

Equivalently, controllability can also be characterized through the controllability Gramian

$$W_T = \int_{t_0}^{t_f} e^{A\tau} BB^\top e^{A^\top \tau} \, d\tau,$$

which is positive definite for some sufficiently large time interval if and only if $(A, B)$ is controllable. The eigenvalues of $W_T$ further quantify the directions that are more or less easily influenced by the inputs.

**Proposition 2** (Controllable Canonical Form). *If the linear system with matrices $(A, B)$ is controllable, there exists an invertible matrix $P$ such that the transformed pair*

$$A_\theta^c = P^{-1}AP, \qquad B_\theta^c = P^{-1}B$$

*takes a block companion form. In the single-input case ($m = 1$), $A_\theta^c$ is the companion matrix*

$$A_\theta^c = \begin{bmatrix} 0 & 1 & 0 & \cdots & 0 \\ 0 & 0 & 1 & \cdots & 0 \\ \vdots & & & \ddots & \vdots \\ 0 & 0 & 0 & \cdots & 1 \\ a_1 & a_2 & a_3 & \cdots & a_N \end{bmatrix}, \qquad B_\theta^c = \begin{bmatrix} 0 \\ 0 \\ \vdots \\ 0 \\ 1 \end{bmatrix},$$

*where $(a_1, \ldots, a_N)$ are real coefficients. In the multi-input case ($m > 1$), $A_\theta^c$ can be written in Brunovský block form with each block corresponding to an input channel. These canonical representations show that every controllable pair is equivalent, up to a similarity transform, to a structured form where controllability is explicit.*

**Definition 5** (State-to-Output Controllability (SOC)). *Consider the linear system $\dot{z}(t) = Az(t) + Bu(t)$ with output $y(t) = Cz(t)$, $y(t) \in \mathbb{R}^p$. The triple $(A, B, C)$ is **state-to-output controllable** if, for any initial state $z(0) = z(t_0)$ and target output $y_T \in \mathbb{R}^p$, there exists a finite horizon and input sequence $u(t) \in \mathbb{R}^m, t \in [t_0, t_f)$, such that $y(t_f) = y_T$.*

Output controllability can be characterized by the output controllability matrix

$$\mathcal{C}_y = \begin{bmatrix} CB & CAB & \cdots & CA^{N-1}B \end{bmatrix} \in \mathbb{R}^{p \times Nm}.$$

The system $(A, B, C)$ is output controllable if and only if $\mathrm{rank}(\mathcal{C}_y) = p$, i.e. the columns of $\mathcal{C}_y$ span the entire output space. When $C = I$, this reduces to the Kalman rank condition for state controllability. Output controllability generalizes state controllability by focusing only on the directions visible through $C$. When $C = I$, the two notions coincide.

If $(A, B)$ is not controllable, there exist directions in the state space that cannot be influenced by any admissible input, which limits the effectiveness of feedback and may render stabilization or trajectory tracking tasks impossible. Even when a system is controllable in the binary sense, the degree of controllability encoded by the Gramian eigenvalues strongly affects input energy requirements and numerical conditioning in control algorithms. For this reason, ensuring or promoting controllability in identified models is essential when the ultimate goal is to use them for control.

## C    CONCEPTUAL OVERVIEW AND CONTRIBUTIONS

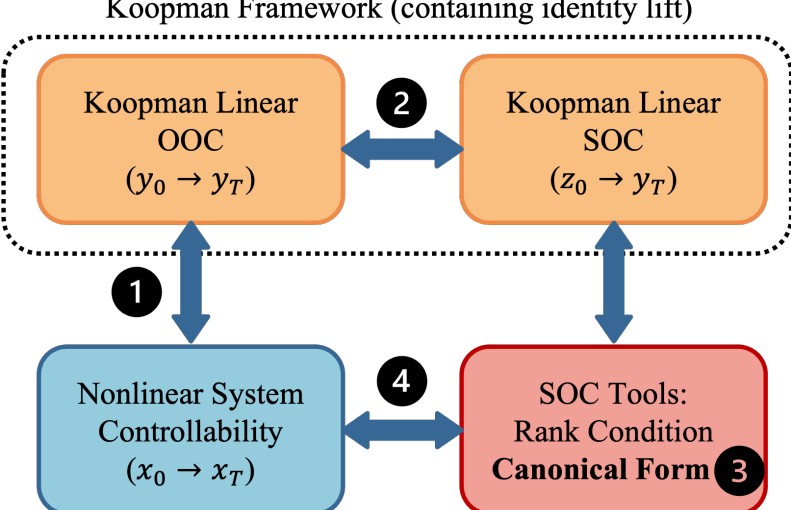

Figure 8: Overview of the relations among controllability notions.

Fig 8 illustrates how different notions of controllability are connected in our framework and our contributions:

① We show that the controllability of the original nonlinear system is equivalent to OOC of Koopman linear representation (in main text).

② In the Koopman linear representation, we prove that OOC coincides with SOC (Lemma 1and Theorem 1, proved in Appendix D).

③ Based on this equivalence, we for the first time introduce a SOC canonical form which can be used for OOC canonical form for Koopman representation (Theorem 2, proved in Appendix  D).

④ Finally, we leverage this canonical form to parameterize the neural Koopman operator, which enables constructing learning-based Koopman representations of the original non-linear system.

# D  PROOF

*Proof of Lemma 1.* We begin by proving that the map $\mathcal{T} : \mathcal{Z} \to \mathbb{R}^n$ with $y = \mathcal{T}z$ is bijective. The map is well-defined from the definition of set $\mathcal{Z}$. For any $z_1, z_2 \in \mathbb{R}^n$, if $y_1 = y_2$, then $z_1 = \phi(y_1) = z_2 = \phi(y_2)$. This indicates that $\mathcal{T}$ is injective. On the other hand, any $y_1 \in \mathcal{Z}$ corresponds to one $z_1 = \phi(y_1)$. This shows that $\mathcal{T}$ is surjective. We conclude that the map $\mathcal{T}$ is bijective.

We then prove sufficiency of the claimant. From the definition, for any $y_0 \in \mathbb{R}^n$, the system can reach any output $y_T \in \mathbb{R}^n$. Consider $z_0 \in \mathcal{Z}$ and $z_T \in \mathcal{Z}$, let $y_0 = Cz_0$ and $y_T = Cz_T$. Given that $\mathcal{T}$ is bijective, it holds that $z_0 = \phi(y_0)$ and $z_T = \phi(y_T)$. From OOC, the system starts from output $y_0$ can reach $y_T$, which indicates that the system starts from state $z_0$ can reach $z_T$. Given that states $z_0$ and $z_T$ are arbitrarily chosen, SOC holds. Necessity of the claimant holds with similar arguments. □

*Proof of Theorem 1.* We begin by proving that the system (3) is SOC on $(\mathcal{Z}, \mathbb{R}^n)$ if and only if $C$ is full-rank. Consider an arbitrary $z_0 \in \mathcal{Z}$ and $y_T \in \mathbb{R}^n$, if there exists an input sequence $u(\cdot) : [t_0, t_f] \to \mathbb{R}^m$ that drives the system from $z_0$ to $y_T$, then we have

$$y_T = C \left( e^{A_\theta(t_f - t_0)} z_0 + \int_{t_0}^{t_f} e^{A_\theta(t_f - t_0 - \tau)} B_\theta u(\tau) d\tau \right) \tag{8}$$

Rearranging the terms and applying Taylor expansion for $e^{A_\theta(t_f - t_0 - \tau)} = \sum_{k=0}^{\infty} \frac{A_\theta(t_f - t_0 - \tau)^k}{k!}$, we have

$$y_T - C e^{A_\theta(t_f - t_0)} z_0 = C \sum_{k=0}^{\infty} A_\theta^k B_\theta \left( \int_{t_0}^{t_f} \frac{(t_f - t_0 - \tau)^k}{k!} u(\tau) d\tau \right) \tag{9}$$

$y_T - C e^{A_\theta(t_f - t_0)} z_0$ spans $\mathbb{R}^n$ because $y_T$ and $z_0$ are independent, and $y_T$ spans $\mathbb{R}^n$. Under this, we deduce that the existence of $u(\tau)$ is equivalent to $C$ is full rank. Using Lemma 1, we conclude that the system is OOC on $\mathbb{R}^n$ if and only if the controllability matrix $C$ is full-rank. □

In the proof of Theorem 1, we use the fact that $y_T - C e^{A_\theta(t_f - t_0)} z_0$ spans $\mathbb{R}^n$ because $y_T$ and $z_0$ are independent, and $y_T$ spans $\mathbb{R}^n$. The subspace $\mathcal{Z}$ does not influence the result. This indicates that for any subspace $\mathcal{H} \subseteq \mathbb{R}^N$ the Koopman linear system is SOC on $(\mathcal{H}, \mathbb{R}^n)$ if and only if it is SOC on $(\mathbb{R}^N, \mathbb{R}^n)$.

*Proof of Theorem 2.* The proof is based on some algebraic calculation. From (5), $A_\theta B_\theta$ and $A_\theta^2 B_\theta$ are given by

$$A_\theta B_\theta = P \begin{bmatrix} 0 \\ 0 \\ \vdots \\ 0 \\ \underbrace{1}_{(n-1)\text{th row}} \\ \vdots \\ \vdots \end{bmatrix} \qquad A_\theta^2 B_\theta = P_\theta \begin{bmatrix} 0 \\ 0 \\ \vdots \\ 0 \\ \underbrace{1}_{(n-2)\text{th row}} \\ \vdots \\ \vdots \end{bmatrix} \tag{10}$$

Using induction, $A_\theta^k B_\theta$ for $1 \le k \le n - 1$ are given by

$$A_\theta^k B_\theta = P_\theta \begin{bmatrix} 0 \\ 0 \\ \vdots \\ 0 \\ \underbrace{1}_{(n-k)\text{th row}} \\ \vdots \\ \vdots \end{bmatrix} \tag{11}$$

The OOC controllability matrix has the following structure

$$
\mathcal{C} = CP
\begin{bmatrix}
0 & 0 & \dots & 0 & \underbrace{1}_{n\text{th column}} & \dots \\
0 & 0 & \dots & \underbrace{1}_{(n-1)\text{th column}} & \dots & \dots \\
\vdots & \vdots & \ddots & \ddots & \ddots & \vdots \\
0 & \underbrace{1}_{(n-1)\text{th row}} & \ddots & \ddots & \ddots & \vdots \\
\underbrace{1}_{(n)\text{th row}} & \dots & \dots & \dots & \dots & \vdots \\
\vdots & \ddots & \ddots & \ddots & \ddots & \vdots
\end{bmatrix}
$$

$$
= P_1
\begin{bmatrix}
0 & 0 & \dots & 0 & \underbrace{1}_{n\text{th column}} & \dots \\
0 & 0 & \dots & \underbrace{1}_{(n-1)\text{th column}} & \dots & \dots \\
\vdots & \vdots & \ddots & \ddots & \ddots & \vdots \\
0 & \underbrace{1}_{(n-1)\text{th row}} & \ddots & \ddots & \ddots & \vdots \\
\underbrace{1}_{(n)\text{th row}} & \dots & \dots & \dots & \dots & \vdots
\end{bmatrix}
\tag{12}
$$

Clearly, $\mathcal{C}$ has full row rank as $P_1$ has full rank and the matrix that multiplies $P_1$ also has full rank. From Theorem 1, system (3) under canonical form (5) is OOC on $\mathbb{R}^n$. $\qquad\square$

# E    EXTENSIONS

## E.1    MULTI-INPUT PARAMETERIZATION

We extend Theorem 2 to multi-input systems with $m \geq 2$. The idea is to consider the Brunovsky canonical form (Luenberger, 2003).

**Theorem 3.** *Consider system* (3) *with $m \geq 2$. Then, the system is OOC on $\mathbb{R}^n$ if and only if there exist matrices $A_\theta^c \in \mathbb{R}^{N \times N}$, $B_\theta^c \in \mathbb{R}^{N \times m}$ and $P \in \mathbb{R}^{N \times N}$, where*

$$
A_\theta^c = \begin{bmatrix} A_1 & 0 & 0 & \ldots & \ldots & 0 \\ f_1 & A_2 & 0 & 0 & \ldots & 0 \\ \vdots & \vdots & \ddots & \ddots & \ddots & \vdots \\ f_2 & f_3 & \ldots & A_m & \ldots & 0 \\ b_1 & b_2 & \ldots & \ldots & \ldots & b_N \\ \vdots & \vdots & \ddots & \ddots & \ddots & \vdots \\ c_1 & c_2 & \ldots & \ldots & \ldots & c_N \end{bmatrix} \quad B_\theta^c = \begin{bmatrix} B_1 & 0 & \ldots & 0 \\ g_1 & B_2 & 0 & \ldots \\ \vdots & \vdots & \ddots & \vdots \\ g_2 & g_3 & \ldots & B_m \\ d_1 & d_2 & \ldots & d_m \\ \vdots & \vdots & \ddots & \vdots \\ e_1 & e_2 & \ldots & e_m \end{bmatrix} \quad P = \mathrm{diag}(P_1, P_2).
$$

(13)

*such that*

$$
A_\theta = P_\theta A_\theta^c P_\theta^{-1} \quad B_\theta = P_\theta B_\theta^c
$$

(14)

*$P_1 \in \mathbb{R}^n, P_2 \in \mathbb{R}^{N-n}$ are both full rank matrices. The blocks $A_i \in \mathbb{R}^{N_i \times N_i}$ and $B_i \in \mathbb{R}^{N_i \times 1}$ are in the forms of*

$$
A_i = \begin{bmatrix} 0 & 1 & 0 & \ldots & \ldots & 0 \\ 0 & 0 & 1 & 0 & \ldots & 0 \\ \vdots & \vdots & \ddots & \ddots & \ddots & \vdots \\ 0 & 0 & \ldots & \ldots & \ldots & 1 \\ a_1 & a_2 & \ldots & \ldots & \ldots & a_{N_i} \end{bmatrix} \quad B_i = \begin{bmatrix} 0 \\ 0 \\ \vdots \\ 0 \\ 1 \end{bmatrix},
$$

(15)

*where $\sum_{i=1}^m N_i = n$, the elements in rows from $n + 1$ to $N$ of both $A_\theta^c$ and $B_\theta^c$ are free.*

*Proof.* To show the structure of the OOC matrix $\mathcal{C}$, we consider using mathematical induction. For $2 < k < N$, suppose that

$$
A_\theta^k B_\theta = P_\theta \begin{bmatrix} A_1^k B_1 & 0 & \ldots & 0 \\ g_1' & A_2^k B_2 & 0 & \ldots \\ \vdots & \vdots & \ddots & \vdots \\ g_2' & g_3' & \ldots & A_m^k B_m \\ d_1' & d_2' & \ldots & d_m' \\ \vdots & \vdots & \ddots & \vdots \\ e_1' & e_2' & \ldots & e_m' \end{bmatrix}
$$

Given that we focus on the structure of the matrix, we use $g'_1, \ldots, e'_m$ to represent some variables or matrices, with a slight abuse of notation. From this, we obtain

$$A_\theta^{k+1} B_\theta = A_\theta \cdot A_\theta^k B_\theta$$

$$= P_\theta \begin{bmatrix} A_1 & 0 & 0 & \ldots & \ldots & 0 \\ f_1 & A_2 & 0 & 0 & \ldots & 0 \\ \vdots & \vdots & \ddots & \ddots & \ddots & \vdots \\ f_2 & f_3 & \ldots & A_m & \ldots & 0 \\ b_1 & b_2 & \ldots & \ldots & \ldots & b_N \\ \vdots & \vdots & \ddots & \ddots & \ddots & \vdots \\ c_1 & c_2 & \ldots & \ldots & \ldots & c_N \end{bmatrix} \begin{bmatrix} A_1^k B_1 & 0 & \ldots & 0 \\ g'_1 & A_2^k B_2 & 0 & \ldots \\ \vdots & \vdots & \ddots & \vdots \\ g'_2 & g'_3 & \ldots & A_m^k B_m \\ d''_1 & d''_2 & \ldots & d'_m \\ \vdots & \vdots & \ddots & \vdots \\ e'_1 & e'_2 & \ldots & e'_m \end{bmatrix}$$

$$\qquad\qquad (16)$$

$$= P_\theta \begin{bmatrix} A_1^{k+1} & 0 & \ldots & 0 \\ g'_1 & A_2^{k+1} B_2 & 0 & \ldots \\ \vdots & \vdots & \ddots & \vdots \\ g'_2 & g'_3 & \ldots & A_m^{k+1} B_m \\ d'_1 & d'_2 & \ldots & d'_m \\ \vdots & \vdots & \ddots & \vdots \\ e'_1 & e'_2 & \ldots & e'_m \end{bmatrix}$$

which satisfies the induction for $k+1$. It an be verified that $A_\theta^2 B_\theta$ also fulfills the structural assumption. Using the result, the OOC matrix $\mathcal{C}$ is given by

$$\mathcal{C} = CP \begin{bmatrix} B_1 & 0 & \ldots & 0 & A_1 B_1 & 0 & \ldots & \ldots & A_1^{N-1} B_1 & 0 & 0 & \ldots \\ g'_1 & B_2 & 0 & \ldots & g'_2 & A_2 B_2 & 0 & \ldots & g'_3 & A_2^{N-1} B_2 & 0 & \ldots \\ \vdots & & & & & \ldots & \ldots & & & & & \vdots \\ g'_4 & \ldots & B_m & \ldots & \ldots & g'_5 & A_m B_m & \ldots & \ldots & g'_6 & A_m^{N-1} B_m & \ldots \\ g'_7 & & & & & \ldots & \ldots & & & & & g'_8 \\ \vdots & & & & & \ldots & \ldots & & & & & \vdots \end{bmatrix}$$

$$= P_1 \begin{bmatrix} B_1 & 0 & \ldots & 0 & A_1 B_1 & 0 & \ldots & \ldots & A_1^{N-1} B_1 & 0 & 0 & \ldots \\ g'_1 & B_2 & 0 & \ldots & g'_2 & A_2 B_2 & 0 & \ldots & g'_3 & A_2^{N-1} B_2 & 0 & \ldots \\ \vdots & & & & & \ldots & \ldots & & & & & \vdots \\ g'_4 & \ldots & B_m & \ldots & \ldots & g'_5 & A_m B_m & \ldots & \ldots & g'_6 & A_m^{N-1} B_m & \ldots \end{bmatrix}$$

$$\qquad\qquad (17)$$

From the companion structure of each $A_i$ and $B_i$, it is known that for each $i$:

$$\begin{bmatrix} B_i & A_i B_i & \ldots & A_i^{N-1} B_i \end{bmatrix} \qquad\qquad (18)$$

is full rank. Inserting zero blocks into the matrix will not change the rank, we then immediately obtain that $\mathcal{C}$ is full rank as well. Using Theorem 1, we conclude that system (3) is OOC. $\qquad\square$

## E.2 SIMPLIFIED PARAMETERIZATION

In the following, we propose another simplified canonical form for single input case.

**Proposition 3.** *Consider system (3) with $m = 1$ (single-input). Then, an OOC canonical form is given by*

$$A_\theta = \begin{bmatrix} \tilde{A} & 0_{n \times (N-n)} \\ 0_{(N-n) \times n} & A' \end{bmatrix} \quad B_\theta = \begin{bmatrix} \tilde{B} \\ B' \end{bmatrix}, \qquad\qquad (19)$$

*where $\tilde{A} \in \mathbb{R}^{n \times n}$ and $\tilde{B} \in \mathbb{R}^{n \times 1}$ are given by*

$$\tilde{A} = \begin{bmatrix} 0 & 1 & 0 & \cdots & 0 \\ 0 & 0 & 1 & \cdots & 0 \\ \vdots & \vdots & \ddots & \ddots & \vdots \\ 0 & 0 & \cdots & 0 & 1 \\ a_0 & a_1 & \cdots & a_{n-2} & a_{n-1} \end{bmatrix} \quad \tilde{B} = \begin{bmatrix} 0 \\ 0 \\ 0 \\ \vdots \\ 1 \end{bmatrix}. \qquad\qquad (20)$$

$A' \in \mathbb{R}^{(N-n)\times(N-n)}$ *and* $B' \in \mathbb{R}^{(N-n)\times 1}$ *are free matrices.*

*Proof.* From (19), the state controllability matrix is given by

$$\mathcal{C}_s = \begin{bmatrix} \tilde{B} & \tilde{A}\tilde{B} & \tilde{A}^2\tilde{B} & \dots & \tilde{A}^{N-1}\tilde{B} \\ B' & A'B' & A'^2B' & \dots & A'^{N-1}B' \end{bmatrix} \tag{21}$$

From Theorem 1, the OOC controllability matrix is given by

$$\mathcal{C} = C\mathcal{C}_s = \begin{bmatrix} \tilde{B} & \tilde{A}\tilde{B} & \tilde{A}^2\tilde{B} & \dots & \tilde{A}^{N-1}\tilde{B} \end{bmatrix}. \tag{22}$$

From (20), the matrix

$$\begin{bmatrix} \tilde{B} & \tilde{A}\tilde{B} & \tilde{A}^2\tilde{B} & \dots & \tilde{A}^{n-1}\tilde{B} \end{bmatrix} \tag{23}$$

is full rank. From the Cayley-Hamilton Theorem, $\mathcal{C}$ is also full rank. $\qquad\square$

This canonical form requires $A_\theta$ to be block diagonal, unlike the one in Theorem 2. This form can be conservative in practice, particularly for the Koopman linear system. This is because the states $x$ and $\phi(x)$ in $z$ have no correlation.

## F    DISCUSSIONS OF LIMITATIONS

### F.1    THE STRUCTURE OF $P_\theta$

The canonical forms in Theorems 2 and 3 reply on an invertible matrix $P_\theta$. This matrix is critical as it provides extra degree of freedom in linearly transforming the coordinates. In these theorems, the matrix $P_\theta$ is constructed as block diagonal $P_\theta = \mathrm{diag}(P_1, P_2)$ because of the OOC structure: it is the first $n$ rows of $[B_\theta^c \quad A_\theta^c B_\theta^c \quad \dots \quad (A_\theta^c)^{N-1} B_\theta]$ that needs to be full rank. Too see why the block diagonal structure is important, we re-evaluate the OOC matrix $\mathcal{C}$ for $m-1$. First, it is clear that

$$\mathcal{C} = CP \underbrace{[B_\theta^c \quad A_\theta^c B_\theta^c \quad \dots \quad (A_\theta^c)^{N-1} B_\theta]}_{:=\mathcal{C}^c}$$

Let matrices $\mathcal{C}_1^c \in \mathbb{R}^{n \times n}, \mathcal{C}_2^c \in \mathbb{R}^{n \times (N-n)}, \mathcal{C}_3^c \in \mathbb{R}^{(N-n) \times n}, \mathcal{C}_4^c \in \mathbb{R}^{(N-n) \times (N-n)}$ be such that

$$\mathcal{C}^c = \begin{bmatrix} \mathcal{C}_1^c & \mathcal{C}_2^c \\ \mathcal{C}_3^c & \mathcal{C}_4^c \end{bmatrix}$$

The companion forms of $A_\theta^c$ and $B_\theta^c$ in Equations (5) and (13) ensure that

$$\mathrm{rank}(\mathcal{C}_1^c) = n \implies \mathrm{rank}\left([B_\theta^c \quad A_\theta^c B_\theta^c \quad \dots \quad (A_\theta^c)^{N-1} B_\theta]\right) \geq n$$

Assuming that $P_\theta$ is full rank but not block diagonal:

$$P = \begin{bmatrix} P_1 & P_2 \\ P_3 & P_4 \end{bmatrix}.$$

Then we have

$$\mathcal{C} = CP\mathcal{C}^c = [P_1 \mathcal{C}_1^c + P_2 \mathcal{C}_2^c \quad P_1 \mathcal{C}_2^c + P_2 \mathcal{C}_4^c.]$$

Although $P_1$, $P_2$ and $\mathcal{C}_1^c$ are ensured to be full rank, $P_1 \mathcal{C}_1^c + P_2 \mathcal{C}_2^c$ is not ensured to be full rank. One *sufficient* condition for $\mathcal{C}$ to be full rank is

$$P_2 = 0, \quad P_1 \text{ is full rank.} \tag{24}$$

For (24) holds and $P_\theta$ is full rank, we adopt a block diagonal structure where both $P_2$ and $P_3$ are set to zero, and $P_1, P_4 \succ 0$. This will unavoidably introduce conservativeness in training.
As we admit that making $P_1 \mathcal{C}_1^c + P_2 \mathcal{C}^c$ full rank by construction is hard, we can still consider constructing a full rank matrix $P_\theta$ with only $P_2 = 0$ to reduce conservativeness. This is left for future investigation.

We also provide some practical insight: any full rank matrix $P$ result in a full rank $\mathcal{C}$ almost surely. If one *fix* a full rank matrix $\mathcal{C}_1^c$ and a not necessarily full rank matrix $\mathcal{C}_2^c$, and randomly construct a matrix $P$ from a continuous distribution. Then, $P_1 \mathcal{C}_1^c + P_2 \mathcal{C}_2^c$ is *almost surely* full rank. This is because the determinant of is a nontrivial polynomial in the entries of $P_1 \mathcal{C}_1^c + P_2 \mathcal{C}_2^c$. The zero set of any nontrivial polynomial has Lebesgue measure zero in $\mathbb{R}^{n^2}$. Since both $P_1$ and $P_2$ are constructed from a continuous joint distribution, it assigns probability zero to Lebesgue-null sets. If $P_1 \mathcal{C}_1^c + P_2 \mathcal{C}_2^2$ is almost surely full rank, then so is $\mathcal{C}$.

### F.2    THE MULTI-INPUT CANONICAL FORM

In Theorem 3, we provides a multi-input canonical form (13). This is obtained from the single-input canonical form within a Brunovsky structure. The whole matrix $A_\theta^c$ consists of $m$ small matrix $A_i \in \mathbb{R}^{N_i \times N_i}$. In linear system theory, the dimensions $N_i$, also termed as *controllability indices*, can be determined via rank checks. More specifically, the index $N_i \geq 3$ for input $i$ is the smallest integer such that

$$\mathrm{rank}([B_\theta \quad A_\theta B_\theta \quad \dots \quad A_\theta^{N_i - 1} B]) - \mathrm{rank}([B_\theta \quad A_\theta B_\theta \quad \dots \quad A_\theta^{N_i - 2} B]) = 1. \tag{25}$$

This condition can be readily verified *a posteriori* for given $A_\theta$ and $B_\theta$. However, the indices are hard to be determined *by construction*, as the correspondence of state and input on the lifted space is unknown. In the experiments, we consider a conservative method that uniformly assigned indices. This is, clearly, not the optimal assignment in general.

# G  EXPERIMENTS

## G.1  DATA COLLECTION

For each environment we simulate continuous-time nonlinear dynamics under random control inputs. At the beginning of each rollout an initial state $x_0$ is drawn uniformly from a prescribed range. The system then evolves according to its ground-truth ODE $\dot{x} = f(x, u)$ while, at each step, the control input $u(t)$ is sampled uniformly from the admissible bounds. Trajectories are recorded at fixed sampling intervals, yielding sequences of length 250 steps.

**Sampling details.**  The integration step size differs across environments: for mountain car and GRN we use $\Delta t = 1.0$ s, while for pendulum and cartpole we use $\Delta t = 0.02$ s. Although our experiments use regularly sampled data, modeling the system in continuous time allows the same framework to handle irregular sampling if required.

**Dataset size.**

- **Mountain Car:** 5000 training trajectories, 1000 test trajectories.
- **Pendulum:** 5000 training trajectories, 1000 test trajectories.
- **Cartpole:** 10000 training trajectories, 1000 test trajectories.
- **GRN:** 10000 training trajectories, 1000 test trajectories.
- **Reacher:** 5000 training trajectories, 1000 test trajectories.
- **Franka:** 5000 training trajectories, 1000 test trajectories.

**Training usage.**  During training, we do not use the entire 250-step rollouts directly. Instead, a sliding window of length 15 is randomly sampled from a trajectory. This increases sample diversity and ensures that the learned models are trained on local temporal contexts.

## G.2  EXPERIMENTS SETTINGS

**Mountain Car dynamics.**  The continuous-time Mountain Car dynamics are:

$$\dot{p} = v,$$
$$\dot{v} = a \cdot P - 0.0025 \cos(3p),$$

where $p$ is the position, $v$ the velocity, $a$ the input action, and $P$ the action scaling coefficient.

Table 3: Hyperparameters for Mountain Car.

| Hyperparameter | Value |
|---|---|
| Batch size | 64 |
| Learning rate | $1 \times 10^{-3}$ |
| Prediction horizon | 15 |
| Dimension of observables $N$ | 16 |
| Number of layers of $\psi$ | 3 |
| Hidden dimension of $\psi$ | 64 |
| Activation function | ReLU |

**Pendulum dynamics.**  The continuous-time Pendulum dynamics are:

$$\dot{\theta} = \omega,$$
$$\dot{\omega} = \frac{3g}{2l} \sin\theta + \frac{3}{ml^2}\tau,$$

where $\theta$ is the angle, $\omega$ the angular velocity, $m$ the mass, $l$ the length, $\tau$ the input torque, and $g$ the gravity constant.

Table 4: Hyperparameters for Pendulum.

| Hyperparameter | Value |
| --- | --- |
| Batch size | 64 |
| Learning rate | $1 \times 10^{-3}$ |
| Prediction horizon | 15 |
| Dimension of observables $N$ | 16 |
| Number of layers of $\psi$ | 3 |
| Hidden dimension of $\psi$ | 64 |
| Activation function | ReLU |

**CartPole dynamics.** The continuous-time CartPole dynamics are:

$$\dot{x} = \dot{x},$$

$$\ddot{x} = \frac{f + l\dot{\theta}^2 \sin\theta}{M} - \frac{l\ddot{\theta}\cos\theta}{M},$$

$$\dot{\theta} = \dot{\theta},$$

$$\ddot{\theta} = \frac{g\sin\theta - \cos\theta\left(\frac{f + l\dot{\theta}^2 \sin\theta}{M}\right)}{l\left(\frac{4}{3} - \frac{m\cos^2\theta}{M}\right)},$$

where $x$ is cart position, $\theta$ the pole angle, $M$ the total mass, $m$ the pole mass, $l$ the pole length, $f$ the input force, and $g$ the gravity constant.

Table 5: Hyperparameters for CartPole.

| Hyperparameter | Value |
| --- | --- |
| Batch size | 64 |
| Learning rate | $1 \times 10^{-3}$ |
| Prediction horizon | 15 |
| Dimension of observables $N$ | 25 |
| Number of layers of $\psi$ | 3 |
| Hidden dimension of $\psi$ | 64 |
| Activation function | ReLU |

**GRN dynamics.** The continuous-time GRN dynamics with 6 states and 3 inputs are:

$$\dot{x}_1 = -\gamma x_1 + a\frac{1}{K + x_6^2} + u_1,$$

$$\dot{x}_2 = -\gamma x_2 + a\frac{1}{K + x_4^2} + u_2,$$

$$\dot{x}_3 = -\gamma x_3 + a\frac{1}{K + x_5^2} + u_3,$$

$$\dot{x}_4 = -cx_4 + \beta x_1,$$

$$\dot{x}_5 = -cx_5 + \beta x_2,$$

$$\dot{x}_6 = -cx_6 + \beta x_3,$$

where $x_1, \ldots, x_6$ are the states, $u_1, u_2, u_3$ are the control inputs, and $\gamma, a, K, c, \beta$ are system parameters.

**Reacher (Mujoco).** Reacher is a two-jointed robot arm. The goal is to move the robot's end effector (called fingertip) close to a target as Figure 9 shown. Note that we should exclude redundant or non-independent states when constructing the OOC canonical form; only the independent degrees of freedom should enter the canonical structure.

Table 6: Hyperparameters for GRN.

| Hyperparameter | Value |
|---|---|
| Batch size | 64 |
| Learning rate | $1 \times 10^{-3}$ |
| Prediction horizon | 15 |
| Dimension of observables $N$ | 30 |
| Number of layers of $\psi$ | 3 |
| Hidden dimension of $\psi$ | 64 |
| Activation function | ReLU |

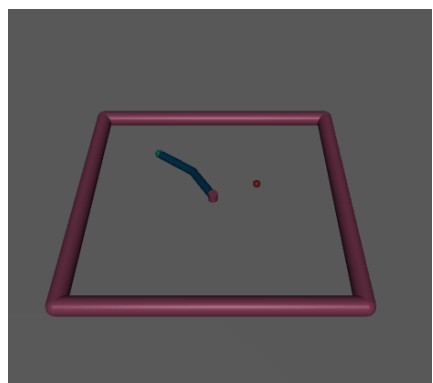

Figure 9: Reacher Environment.

**Franka (7 DoF Manipulator).** Franka is a 7 DoF robotic manipulator. The goal is to move the robot's end effector (called fingertip) close to a target. The environment setting in our experiments is the same as provided in Shi & Meng (2022). Note that we should exclude redundant or non-independent states when constructing the OOC canonical form; only the independent degrees of freedom should enter the canonical structure.

## G.3 ADDITIONAL RESULTS

**Mountain Car** The comparison of prediction performance on Mountain Car Environment with 1% training dataset are shown in Figure 10

**Pendulum** The comparison of control performance on Pendulum environment is shown is Figure 11

**Cartpole** The comparison of prediction and control performance on Cartpole environment is shown in Figure 12 and Figure 13.

**GRN** The results of prediction on GRN environment by our method is shown in Figure 16.

**Reacher** The results of prediction on Reacher environment over different amount of training by DKO and our methos are shown in in Table 9. Also, the training loss versus step is shown in Figure 15a. Our losses decrease rapidly and stabilize at a lower level, whereas DKO starts with large errors and converges slowly.

**Franka** The results of prediction on Reacher environment over different amount of training by DKO and our methos are shown in in Table 9. Also, the training loss versus step is shown in Figure 15b. The results of prediction on GRN environment by our method is shown in Figure 16. It can be found that our method learns faster with fewer data needed to achieve satisfying prediction performance.

Table 7: Hyperparameters for Reacher.

| Hyperparameter | Value |
|---|---|
| Batch size | 64 |
| Learning rate | $1 \times 10^{-2}$ |
| Prediction horizon | 15 |
| Dimension of observables $N$ | 16 |
| Number of layers of $\psi$ | 3 |
| Hidden dimension of $\psi$ | 64 |
| Activation function | ReLU |

Table 8: Hyperparameters for Franka.

| Hyperparameter | Value |
|---|---|
| Batch size | 64 |
| Learning rate | $1 \times 10^{-3}$ |
| Prediction horizon | 15 |
| Dimension of observables $N$ | 16 |
| Number of layers of $\psi$ | 3 |
| Hidden dimension of $\psi$ | 64 |
| Activation function | ReLU |

### G.4 ROLE OF CONTROLLABILITY LOSS

Suitable penalty on controllability loss will also contribute to accuracy. However, too large penalty makes the model must trade off matching the true dynamics versus enlarging controllability, then may damage prediction accuracy. Although MPC can compensate for prediction inaccuracies to some extent, poor prediction quality results in higher control effort in the end. The experiments on Pendulum with different weights on gramian loss (0.005 and 0.05) are shown in Figure 17.

### G.5 ADDING ENCODER FOR INPUT

Adding an input encoder as shown in Figure 18 effectively introduces a nonlinear dependence of the control signal on the state, since the input to the lifted dynamics becomes $\hat{u} = g_\theta \odot u$ rather than the raw control signal. Formally, this representation takes the form:

$$\dot{z}(t) = A_\theta z(t) + B_\theta \hat{u}(t), \quad z(t) = \phi_\theta(x(t)),$$
$$\hat{x}(t) = y(t) = Cz(t), \quad \hat{u}(t) = g_\theta(\hat{x}(t)) \odot u(t)$$

Although $g$ may depend on the state, the surrogate system can still be regarded as linear in its effective input $\hat{u}$. Consequently, the controllability analysis remains unchanged: as long as the encoder does not map all admissible inputs to zero, the pair $(A, B)$ governs reachability in exactly the same way as in the standard formulation without encoding. In this sense, the encoder enriches the expressiveness of the Koopman representation without undermining the controllability guarantees.

The prediction performance of our framework with input encoder is shown in Table 10. It can be found the prediction is even more accurate. However, the reason why we still choose the framework without input encoder is: in this case the effective input becomes $\hat{u} = g(\hat{x}) \odot u$, leading to a bilinear system $\dot{z} = Az + B(g(Cz) \odot u)$. This breaks the standard linear structure used in MPC (which could be our future research direction); one could optimize over $\hat{u}$, but then the original control $u$ is no longer explicit, making constraints and interpretation of the optimization objective difficult. We therefore focus on the direct-input setting, which integrates more naturally with control design.

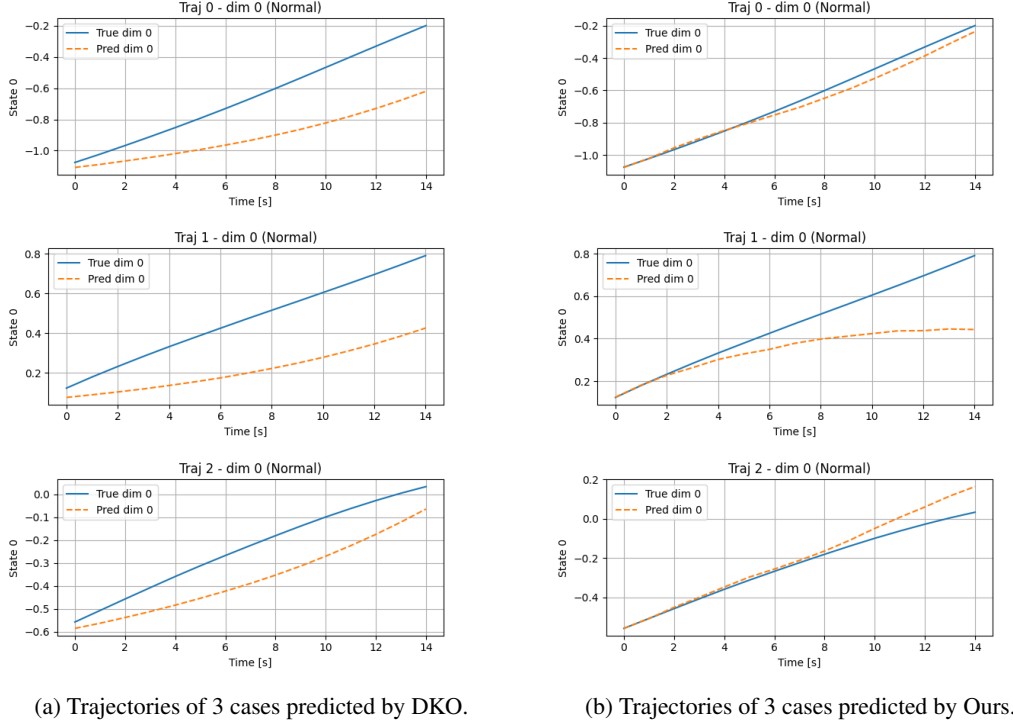

(a) Trajectories of 3 cases predicted by DKO.    (b) Trajectories of 3 cases predicted by Ours.

Figure 10: Comparison of Prediction Performance on Mountain Car Environment with 1% training dataset

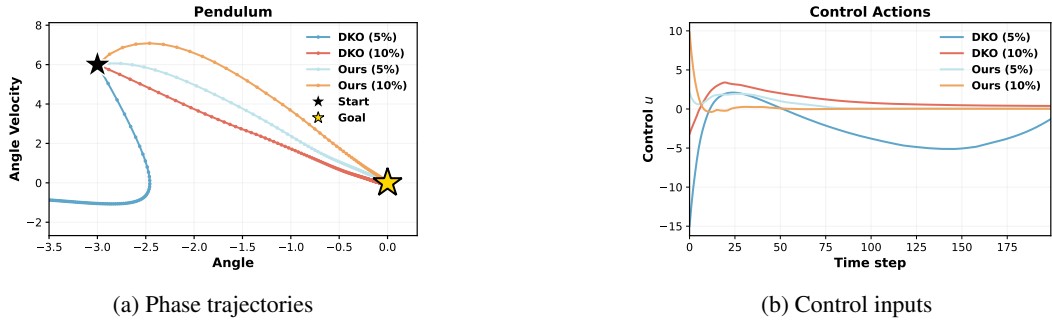

(a) Phase trajectories    (b) Control inputs

Figure 11: Comparison of Pendulum experiments. (a) trajectories in angle–velocity space; (b) corresponding control sequences.

Table 9: Comparison of prediction error (MSE) across environments with varying fractions of training data for Reacher and Franka.

| Environment | Method | 1% | 5% | 10% | 30% | 50% | 100% |
|---|---|---|---|---|---|---|---|
| Reacher | DKO | > 5 | 3.47 | 1.55 | 0.5613 | 0.2572 | 0.0194 |
|  | Ours |  | **0.0199** | **0.0135** | **0.0039** | $\leq \mathbf{1 \times 10^{-4}}$ | |
| Franka | DKO | 0.0453 | 0.022 | 0.0068 | $10^{-4}$ | $\leq 1 \times 10^{-5}$ | |
|  | Ours | **0.039** | **0.00064** | | $\leq \mathbf{1 \times 10^{-5}}$ | | |

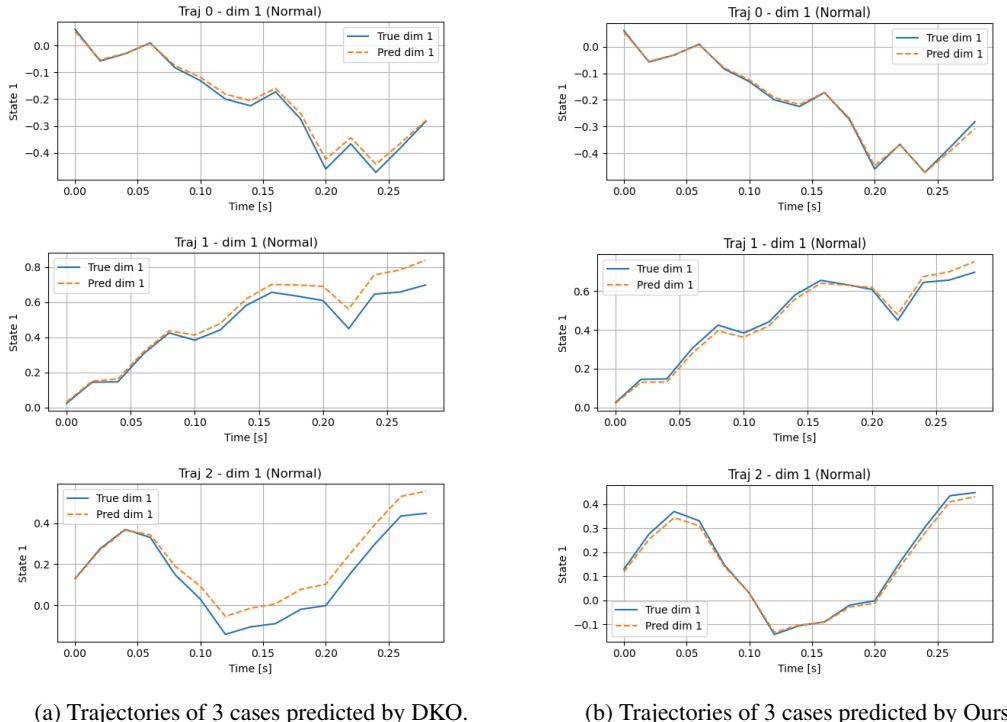

(a) Trajectories of 3 cases predicted by DKO.   (b) Trajectories of 3 cases predicted by Ours.

Figure 12: Comparison of Prediction Performance on Cartpole Environment with 10% training dataset

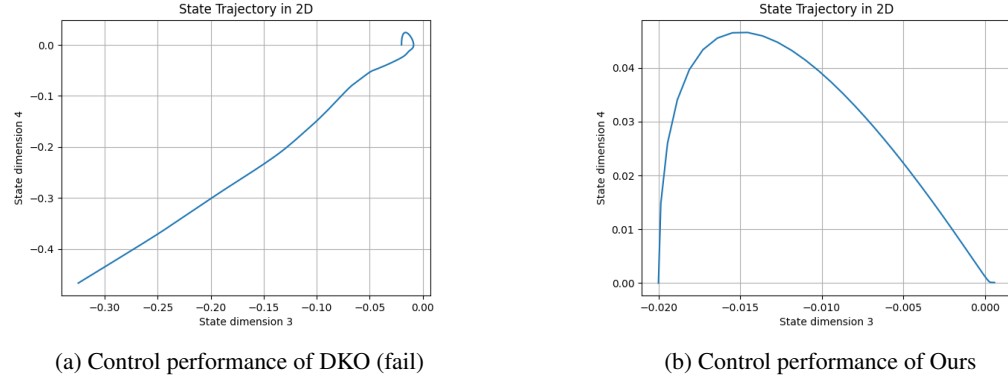

(a) Control performance of DKO (fail)     (b) Control performance of Ours

Figure 13: Comparison of Control performance on Cartpole with 30% training dataset

Table 10: Comparison of prediction error (MSE) across environments with varying fractions of training data. Our method shows clear advantages under limited training data, highlighting its data efficiency.

| Environment | Method | 1% | 5% | 10% |
|---|---|---|---|---|
| Mountain Car | w/o Input Encoder | 0.0032 | 0.00019 | 0.00011 |
| | w/ Input Encoder | **0.00087** | **0.00018** | 0.00011 |
| Pendulum | w/o Input Encoder | 0.3747 | 0.0318 | 0.0114 |
| | w/ Input Encoder | **0.1338** | **0.0067** | **0.0065** |

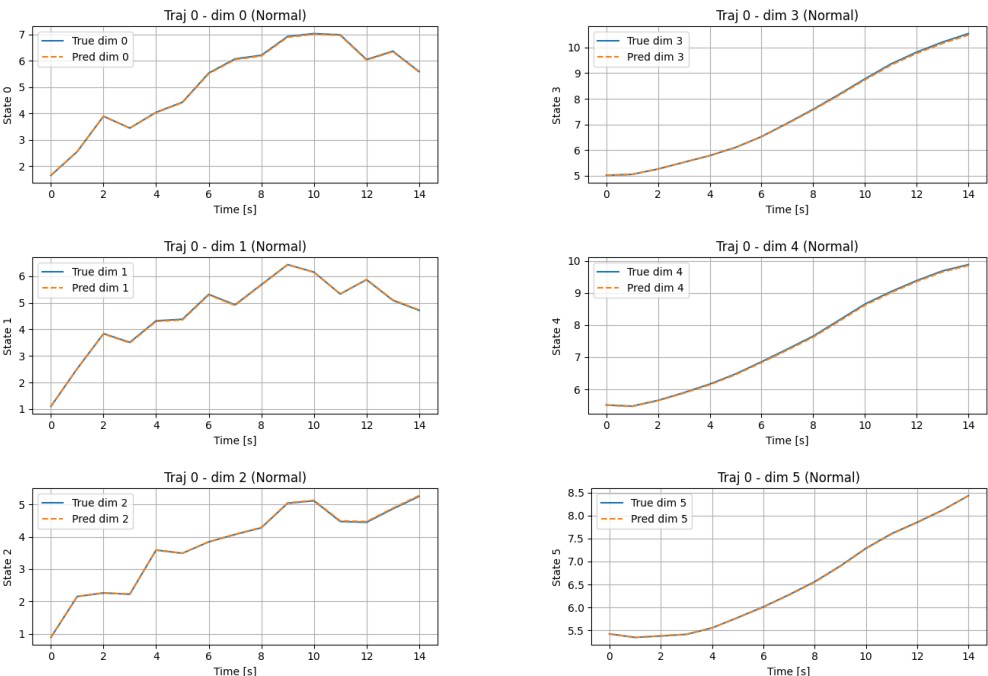

Figure 14: Prediction Performance on GRN Environment

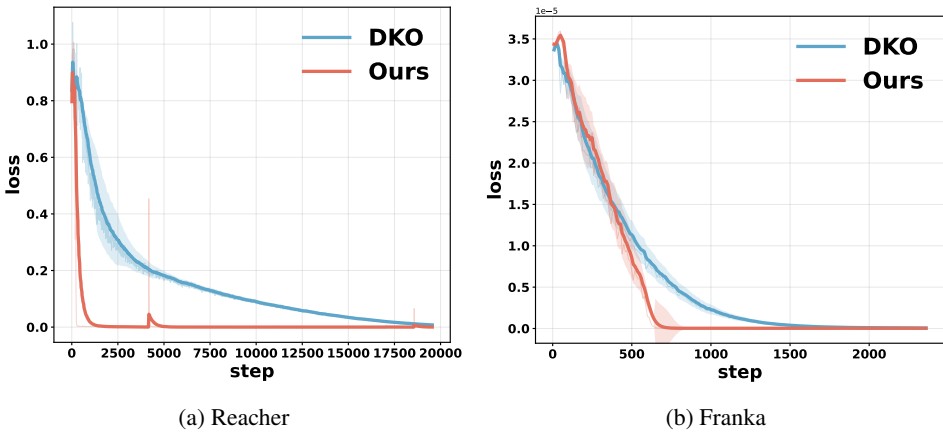

(a) Reacher

(b) Franka

Figure 15: Loss versus training step on Reacher and Franka. Our method converges faster than DKO.

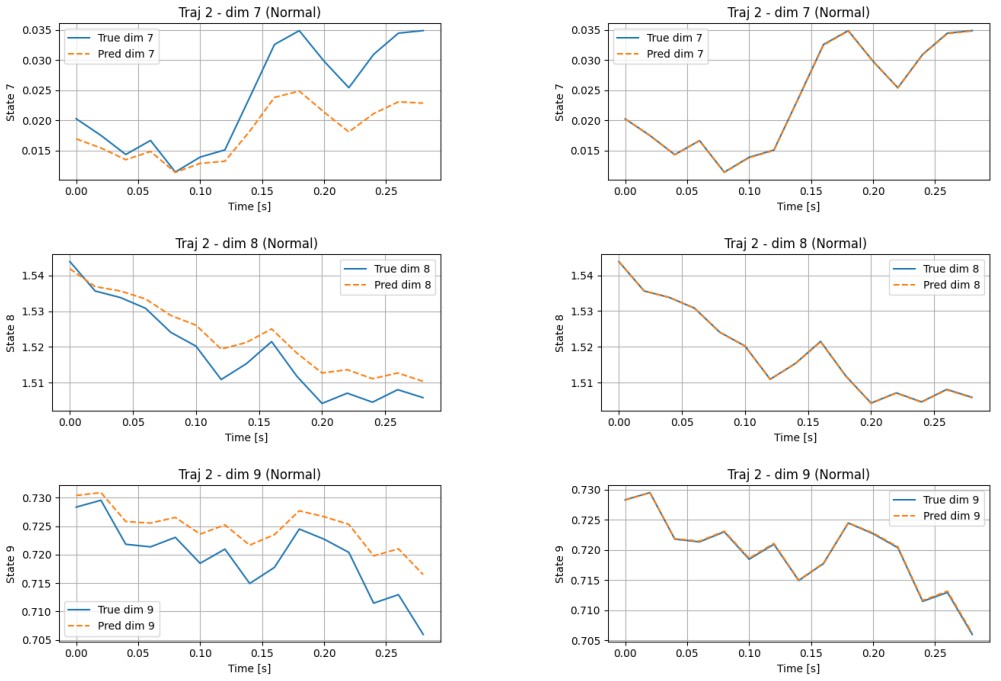

(a) Trajectories of end effector predicted by DKO.  (b) Trajectories of end effector predicted by Ours.

Figure 16: Prediction Performance (end effector position) on Franka Environment with 10% training dataset

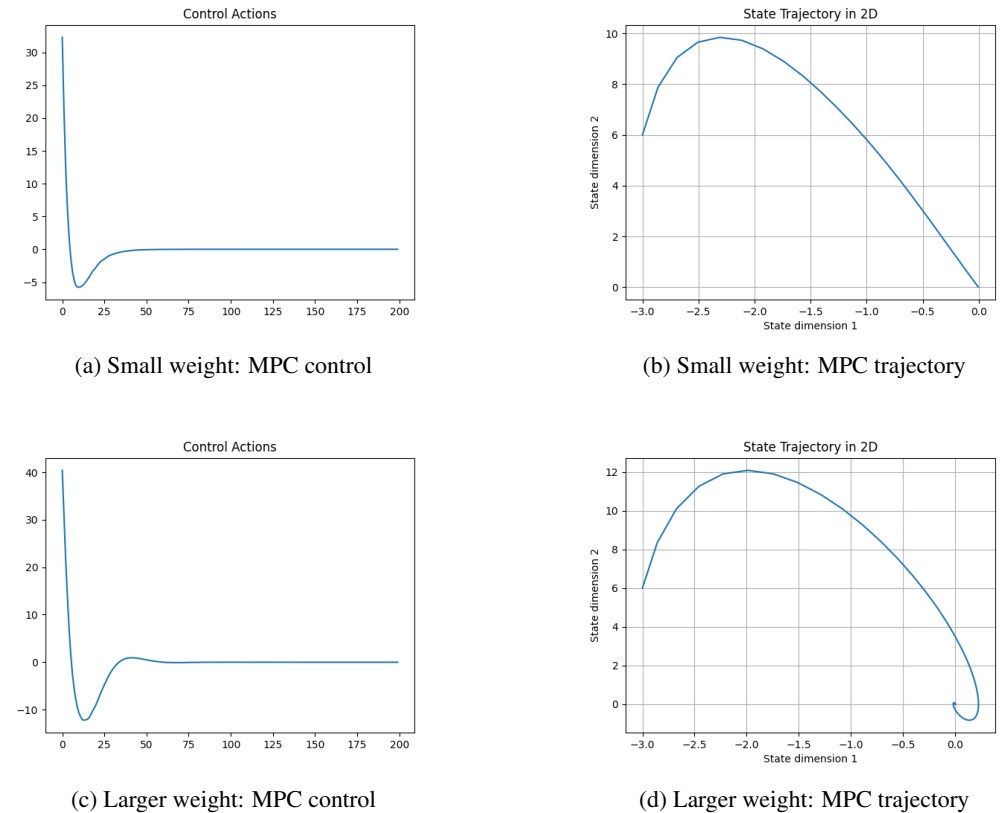

(a) Small weight: MPC control

(b) Small weight: MPC trajectory

(c) Larger weight: MPC control

(d) Larger weight: MPC trajectory

Figure 17: Comparison of MPC performance under small and larger weights settings. The larger weight setting shows larger eigenvalues but poorer predictive control (larger energy use and overshoot in trajectory).

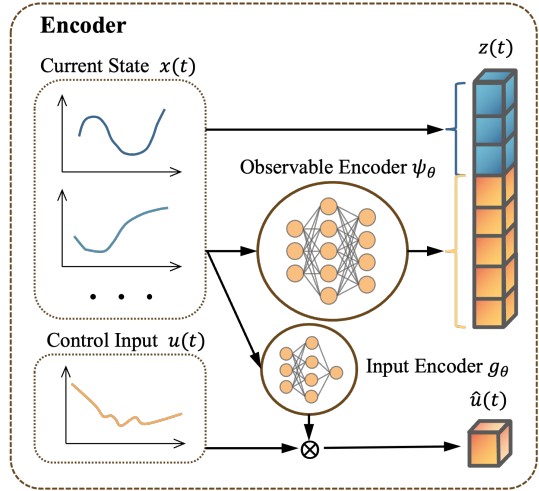

Figure 18: With Input Encoder

