# OpenReview forum: "Learning Koopman Representations with Controllability Guarantees"
_ICLR.cc/2026/Conference — ICLR 2026 Poster_

### Official Review · Reviewer_dLx9 · 2025-10-31

**Soundness:** 3
**Presentation:** 3
**Contribution:** 3
**Rating:** 6
**Confidence:** 4

**Summary:**

This paper proposes a framework for learning Koopman representations of nonlinear control systems that are guaranteed to be controllable. The authors derive conditions linking output-to-output controllability (OOC) and standard state-output controllability (SOC), and introduce a canonical parameterization of the latent linear dynamics matrices A_\theta, B_\theta that ensures controllability by construction. Additionally, a controllability Gramian regularizer is used to improve the conditioning (“well-controllable” property). The learned model can be used directly in model predictive control (MPC). Experiments on benchmark systems demonstrate improved sample efficiency and control performance compared to standard Deep Koopman (DKO) and MLP-based models.

**Strengths:**

S1. Embedding controllability directly into the Koopman parameterization is novel. Prior deep Koopman and neural ODE models treat controllability as an afterthought or soft constraint. The proposed canonical form and similarity transform provide an elegant, principled way to enforce this property.

S2. The proofs linking OOC↔SOC and the use of the finite-time output controllability Gramian are mathematically well-grounded and consistent with classical control theory. This makes the paper attractive to both ML and control communities.

S3. The ability to plug the learned model directly into linear MPC is a significant engineering advantage. The demonstration that the approach maintains performance under limited data conditions is also persuasive.

**Weaknesses:**

W1. There is limited novelty relative to the existing Koopman literature. While embedding controllability guarantees is new, the core architecture—learning a linear operator in latent space via encoder + linear dynamics + decoder—is largely inherited from standard deep Koopman frameworks. The contribution thus appears incremental in form but conceptual in framing. The authors should clarify in what sense their controllability parameterization extends beyond “structured Koopman learning” or prior work on “control-aware embeddings.”

W2. Controllability parameterization may constrain expressive power. The use of a canonical controllable form plus similarity transform P_\theta guarantees controllability but may significantly restrict the learned dynamics manifold. It is unclear whether the proposed structure can approximate arbitrary Koopman operators as the dimension of z increases, or whether expressiveness is traded off for structural rigidity.

W3. There seems to be no stability guarantees. The method ensures controllability but says little about the stability of the learned system -  an equally critical property for control deployment. Without constraints on the eigenvalues of A_\theta, the learned linear system may be unstable, making MPC optimization difficult or ill-posed.

W4. All benchmarks are relatively small and well-known toy systems (Pendulum, CartPole, etc.). While these are standard, they do not convincingly demonstrate the method’s scalability or robustness to real-world noise and unmodelled dynamics.

W5. Computational cost and training stability are unclear. The paper lacks analysis of training complexity, numerical conditioning, or the computational overhead introduced by the Gramian regularizer (which involves integrating matrix exponentials).

W6. The connection to Koopman theory is not fully justified. Although the paper uses the Koopman terminology, the approach essentially learns a linear latent-space model via a neural encoder. There is limited evidence that this latent space truly captures Koopman-invariant subspaces or observables.

**Questions:**

Q1. The paper constrains A_\theta, B_\theta through a controllable canonical form plus similarity transform P_\theta. Does this restriction reduce the expressive power of the learned Koopman operator? Please clarify whether your parameterization still provides a universal approximation property for arbitrary nonlinear dynamics in the lifted space, or discuss the trade-off between controllability enforcement and representational flexibility.

Q2. The current method enforces controllability but not stability. Have you observed instability in the learned latent dynamics (e.g., exploding eigenvalues of A_\theta)? It would be useful to regularize eigenvalues of A_\theta or provide evidence that the MPC formulation mitigates instability during rollout.

Q3. The Gramian term involves computing W_T^y. How is this computed efficiently during training, and how does it scale with the latent dimension N? It would be helpful to include complexity analysis or ablation to show that the regularizer does not dominate training time.

Q4. The framework is called “Koopman,” but it remains unclear whether the learned observables correspond to Koopman-invariant subspaces or simply linear latent embeddings. Can you empirically or theoretically justify that \varphi_\theta(x) approximates Koopman eigenfunctions or invariant coordinates? Spectral or mode analysis (e.g., eigenvalue comparison with known systems) would strengthen the Koopman interpretation.

Q5. Can your framework scale to higher-dimensional or partially observed systems (e.g., PDEs, power network, or soft-robot system)? Discuss computational limits and whether encoder-decoder architectures can handle such cases without losing controllability guarantees.

Q6. The paper does not compare against recent Koopman or geometric control approaches (e.g., KEEC) that emphasize structural invariance. What are the pros and cons of controllability-based representations and equivariance-based representations?

Q7. Parameters like the Gramian regularization weight and time horizon T may strongly affect results. How sensitive is performance to these settings? Suggest including an ablation or at least a qualitative discussion on how these hyperparameters influence both controllability conditioning and prediction accuracy.

Q8. The paper claims the learned model can be “directly integrated” with MPC. Was this tested in a closed-loop simulation with constraints? Suggest presenting control performance metrics (tracking error, energy use, robustness) under realistic MPC settings to demonstrate deployability.

---

> ### Author Response · Authors · 2025-11-25
> **Response to Reviewer dLx9**
>
> ### Summary
>
> Thank you for the very professional and thoughtful review. From your comments, it is clear that you have a deep understanding of the Koopman field. We found many of your insights highly valuable, and they align closely with our own research findings. We would nevertheless like to clarify a few points you raised, particularly regarding our contributions.
>
>
> ### Response to Weaknesses
>
> **Comment:** *There is limited novelty relative to the existing Koopman literature. While embedding controllability guarantees is new, the core architecture—learning a linear operator in latent space via encoder + linear dynamics + decoder—is largely inherited from standard deep Koopman frameworks. The contribution thus appears incremental in form but conceptual in framing. The authors should clarify in what sense their controllability parameterization extends beyond "structured Koopman learning" or prior work on "control-aware embeddings."*
>
> **Response:** Thank you for your comment, and we appreciate your acknowledgment of our contribution on embedding controllability. The reviewer is absolutely correct that our work is built upon the classical deep Koopman frameworks. However, we argue that our proposed control aware structure is original and significant, addressing a long standing open question: how to make the Koopman model applicable for downstream control.
>
> As pointed out by [1], ensuring State to State Controllability (SSC) of the Koopman system is generally impossible even when the original nonlinear system is controllable. This limitation is clear, since the dimension of the control input remains unchanged in the lifted space while the state dimension increases. The lack of SSC restricts the applicability of many optimization based control methods, such as MPC and LQR, without substantial reformulation. Subsequent works, such as [2], proposed complex infinite dimensional extensions of the Lie algebra rank condition to verify controllability. However, this condition is difficult to verify and even more challenging to enforce by design. More recently, the authors of [3] provided sufficient conditions on the surjectiveness of observables and invariance of dictionaries for controllability preservation. Yet they did not demonstrate how to ensure these conditions by design, nor were any of the proposed conditions validated in their experiments.
>
> To address this problem, we adopt a different perspective based on State to Output Controllability (SOC) and the recently proposed concept of Output to Output Controllability (OOC) [4], rather than SSC of the Koopman system. The motivation is that *OOC of the Koopman system is equivalent to SSC of the corresponding learned nonlinear system*, which makes it essential for control design. To overcome the difficulty of enforcing OOC in the learned Koopman system, we establish the equivalence between SOC and OOC for the Koopman linear system and then propose a novel OOC canonical parameterization. Our framework is the first in this field to treat controllability as a structural prior for Koopman learning, and also the first to ensure *controllability by design* in learning control aware embeddings.
>
> [1] D. Goswami and D. A. Paley, “Bilinearization, reachability, and optimal control of control-affine nonlinear systems: A Koopman spectral approach,” IEEE Transactions on Automatic Control, vol. 67, no. 6, pp. 2715–2728, 2021.
>
> [2] W. Zhang and J.-S. Li, “Koopman bilinearization of nonlinear control systems,” arXiv preprint arXiv:2211.07112, 2022.
>
> [3] Choi, Joonwon, Minhyun Cho, Hyunsang Park, Vishnu Vijay, and Inseok Hwang. “On The Controllability Preservation of Koopman Bilinear Surrogate Model.” In 2024 IEEE 63rd Conference on Decision and Control (CDC), pp. 3457–3462. IEEE, 2024.
>
> [4] Danhane, Baparou, Jérôme Lohéac, and Marc Jungers. “Characterizations of output controllability for LTI systems.” Automatica 154 (2023): 111104.
>
> ---
>
> **Comment:** *There seems to be no stability guarantees. The method ensures controllability but says little about the stability of the learned system – an equally critical property for control deployment. Without constraints on the eigenvalues of $A_{\theta}$, the learned linear system may be unstable, making MPC optimization difficult or ill-posed.*
>
> **Response:** Thank you for this interesting point. The reason we focus on controllability is that the stability of the closed loop system is ensured by the control design for ***non autonomous*** systems. Stability of the uncontrolled system itself is not required when using MPC, and in practice very few nonlinear dynamical systems are stable in open loop. Investigating stability or passivity through structural constraints in learning ***autonomous*** Koopman models is indeed another interesting direction.
>
> ---

---

> ### Author Response · Authors · 2025-11-25
>
> ---
>
> **Comment:** *Controllability parameterization may constrain expressive power. The use of a canonical controllable form plus similarity transform $P_\theta$ guarantees controllability but may significantly restrict the learned dynamics manifold. It is unclear whether the proposed structure can approximate arbitrary Koopman operators as the dimension of $z$ increases, or whether expressiveness is traded off for structural rigidity.*
>
> **Response:** The reviewer is absolutely correct in this observation. In theory, the canonical parameterization may constrain expressive power. This arises mainly from the structure of the similarity transformation matrix $P_θ$, which is block diagonal and symmetric. We already discussed this issue in Appendix F of the paper. The key point in that discussion is a practical insight: any full rank matrix $P_θ$ yields a full rank OOC controllability matrix $C$ almost surely. Removing the structural restrictions on $P_θ$ can therefore improve the expressive capacity of the parameterization.
>
> In our experiments, we observe that even with a block diagonal and symmetric similarity transformation matrix $P_θ$, the proposed canonical parameterization produces Koopman models that exhibit higher fidelity than plain deep Koopman methods and MLPs. With more data, all methods achieve similar learning precision, which further demonstrates the expressive capability of our approach.
>
> A rigorous universal expressivity analysis of our method is an important direction and will be explored in future work.
>
> ---
>
> **Comment:** *All benchmarks are relatively small and well-known toy systems (Pendulum, CartPole, etc.). While these are standard, they do not convincingly demonstrate the method’s scalability or robustness to real-world noise and unmodelled dynamics.*
>
> **Response:** We appreciate the reviewer’s suggestion. Our experimental evaluation includes standard control benchmarks widely used in the Koopman literature (Mountain Car, Inverted Pendulum, and CartPole), as well as the gene regulatory network (GRN) system, which is a less commonly explored domain in Koopman based work. For context, the recent ICLR 2025 paper [1], as cited in our manuscript, is one of the few works that also includes the GRN system and similarly evaluates on these benchmark tasks without extending to more complex robotic systems. Given this context, we believe our evaluation provides a fair comparison on the benchmarks that are established and widely used in the field.
>
> We understand that the reviewer has higher expectations for our method, and we share the enthusiasm about testing our framework on more complex systems. To address this, we have conducted additional experiments on two more complex robotic systems: Reacher from the MuJoCo suite and the Franka robot with 7 degrees of freedom [2].
>
> A summary of the new results is:
>
> **Table: Comparison of prediction error across environments with varying fractions of training data.**
>
> | Environment | Method | 1%             | 5%       | 10%      | 30%      | 50%                 | 100%                |
> |-------------|--------|----------------|----------|----------|----------|----------------------|----------------------|
> | **Reacher** | DKO    | >5             | 3.47     | 1.55     | 0.5613   | 0.2572               | 0.0194               |
> |             | Ours   | >5             | **0.0199**   | **0.0135**   | **0.0039**   | **≤1×10⁻⁴**              | **≤1×10⁻⁴**              |
> | **Franka**  | DKO    | 0.0453         | 0.022    | 0.0068   | 1×10⁻⁴   | ≤1×10⁻⁵              | ≤1×10⁻⁵              |
> |             | Ours   | **0.039**          | **0.00064**  | **≤1×10⁻⁵**  | **≤1×10⁻⁵**  | **≤1×10⁻⁵**              | **≤1×10⁻⁵**              |
>
> We have put this table and together with figures on the prediction performance and training loss in Appendix G in out revised manuscript. These results show that our controllability and observability constraints remain effective as system complexity increases. Together with our existing evaluation across mechanical control tasks and biological systems, this provides a comprehensive assessment of our framework’s applicability within the scope where Koopman operator methods are currently viable.
>
> [1] Li, Zhaoyang, Minghao Han, and Xunyuan Yin. "MamKO: Mamba-based koopman operator for modeling and predictive control." The Thirteenth International Conference on Learning Representations. 2025.
>
> [2] Shi, Haojie, and Max Q-H. Meng. "Deep Koopman operator with control for nonlinear systems." IEEE Robotics and Automation Letters 7.3 (2022): 7700-7707.
>
> ---

---

> > ### Author Response · Authors · 2025-11-25
> >
> > **Comment:** *Computational cost and training stability are unclear. The paper lacks analysis of training complexity, numerical conditioning, or the computational overhead introduced by the Gramian regularizer (which involves integrating matrix exponentials).*
> >
> > **Response:** We thank the reviewer for this good question. There are many ways to avoid integrating matrix exponentials. While the Gramian
> > $$W_c(T) = ∫_0^T e^{At} B B^\top e^{A^\top t} dt$$
> > can be written formally in terms of matrix exponentials, there exist multiple standard methods to compute it without ever evaluating $e^{At}$.
> >
> > Our implementation uses a *coupled ODE system*: we evolve the state transition matrix $$\dot{Φ}(t) = A Φ(t),  Φ(0) = I,$$ and simultaneously integrate  $$\dot{Q}(t) = Φ(t) B B^\top Φ(t)^\top,  Q(0) = 0,$$ so that $W_c(T) = Q(T)$. Alternatively, one can directly integrate the Lyapunov differential equation $$\dot{W}_c(t) = A W_c(t) + W_c(t) A^\top + B B^\top,  W_c(0) = 0.$$
> >
> > Both approaches use only standard matrix multiplications at each integration step, avoiding matrix exponential computations entirely.
> >
> > **Empirical overhead is modest.**
> > Using the pendulum experiment ($n_z = 18$) as an example for ablation, and with our problem size and batch size (64), this contributes only a modest portion of the overall cost. Empirically, enabling the Gramian regularizer increases wall clock training time by a factor of approximately from 1.2 to 1.3 times compared to training without it (see the table for detailed timings across experiments). The table also shows that training remains fast even when the Gramian regularizer is enabled.
> >
> > | Experiment | Gramian | 10% | 30% | 50% |
> > |-----------|---------|-----|-----|-----|
> > | Pendulum  | w/o     | 7s  | 23s | 38s |
> > |           | w/      | 9s  | 29s | 48s |
> >
> > **Overall training time remains practical.**
> > As shown in this table, training remains fast even when the Gramian is used. Crucially, our framework achieves good predictive accuracy with fewer data samples and training iterations compared to unconstrained approaches, as shown in Table 1 in our manuscript. This sample efficiency, enabled by the control theoretic constraints, means that despite the approximately 1.2 times per iteration overhead, the total wall clock time required to reach a given level of performance remains competitive or even favorable. The modest computational cost of the Gramian term is therefore offset by faster convergence and reduced data requirements.
> >
> > **Further acceleration is possible.** If desired, the Gramian can be computed without any numerical integration using Van Loan’s block matrix exponential formula [1]. Defining
> > $$
> > M =
> > \begin{bmatrix}
> > A & BB^\top \\\\
> > 0 & -A^\top
> > \end{bmatrix}
> > $$
> >
> > $$
> > e^{MT} =
> > \begin{bmatrix}
> > \Phi_{11}(T) & \Phi_{12}(T) \\\\
> > 0            & \Phi_{22}(T)
> > \end{bmatrix}
> > $$
> > we have $W_c(T) = Φ_{12}(T) Φ_{11}(T)^\top$, requiring only a single $(2 n_z) × (2 n_z)$ matrix exponential per update rather than iterative integration.
> >
> > [1] Van Loan, Charles. "Computing integrals involving the matrix exponential." IEEE Transactions on Automatic Control 23.3 (1978): 395–404.
> >
> > ---

---

> ### Author Response · Authors · 2025-11-25
>
> ---
>
> **Comment:** *The connection to Koopman theory is not fully justified. Although the paper uses the Koopman terminology, the approach essentially learns a linear latent-space model via a neural encoder. There is limited evidence that this latent space truly captures Koopman-invariant subspaces or observables.*
>
> **Response:** Thank you, this is an important and interesting question. Some dictionaries used in EDMD methods belong to a space of observables for which $ψ(x_n)$ is $\mathcal{K}$-invariant for all $n$, such as polynomial functions. However, many widely used dictionaries in EDMD still do not form invariant spaces, such as RBF dictionaries, although they are widely used because of their strong expressive power. As noted in [1], identifying regression models based on nonlinear measurements *“will generally result in closure issues, as there is no guarantee that these measurements form a Koopman invariant subspace.”* The authors further explain that *“eigenfunctions are guaranteed to span an invariant subspace, and the Koopman operator will yield a matrix when restricted to this subspace, but in practice Koopman eigenfunctions may be more difficult to obtain than the solution of the Koopman operator.”* They conclude that *“the challenge of identifying and representing Koopman eigenfunctions provides strong motivation for the use of powerful emerging deep learning methods.”*
>
> Deep Koopman frameworks, including ours, which utilize deep neural networks as observable functions, generally enforce equivariance implicitly through the loss function. Several prior works have discussed the difficulty of preserving invariance. For example, [2] pointed out that *“if the network has hybrid or bifurcative behaviour, e.g. a switching network, then the dictionary functions may have the capacity to model multiple invariant subspaces of the Koopman operator simultaneously.”* Subsequent works [3] and [4] conducted more thorough analyses, showing that heterogeneous mixtures of dictionary functions can approximate subspace invariance for Koopman operators. We will include a discussion of these works and this issue in the revised version.
>
> We also note that our method does not depend on any specific choice of dictionary functions. This is because the canonical parameterization operates on the state and input matrices $A_θ$ and $B_θ$. Although our implementation uses deep learning, the method can be applied equally well with other approaches where the dictionary functions correspond to Koopman-invariant subspaces.
>
> [1] Lusch, Bethany, J. Nathan Kutz, and Steven L. Brunton. “Deep learning for universal linear embeddings of nonlinear dynamics.” Nature Communications 9.1 (2018): 4950.
>
> [2] Yeung, Enoch, Soumya Kundu, and Nathan Hodas. “Learning deep neural network representations for Koopman operators of nonlinear dynamical systems.” 2019 American Control Conference (ACC). IEEE, 2019.
>
> [3] Johnson, Charles A., Shara Balakrishnan, and Enoch Yeung. “Learning Invariant Subspaces of Koopman Operators–Part 1: A Methodology for Demonstrating a Dictionary’s Approximate Subspace Invariance.” arXiv preprint arXiv:2212.07358 (2022).
>
> [4] Johnson, Charles A., Shara Balakrishnan, and Enoch Yeung. “Heterogeneous Mixtures of Dictionary Functions to Approximate Subspace Invariance in Koopman Operators: Why Deep Koopman Operators Works.” Journal of Nonlinear Science 35.4 (2025): 68.
>
> ---
> ### Answers to Questions
>
> **Question:** *The paper constrains $A_θ$, $B_θ$ through a controllable canonical form plus similarity transform $P_θ$. Does this restriction reduce the expressive power of the learned Koopman operator? Please clarify whether your parameterization still provides a universal approximation property in the lifted space, or discuss the trade-off between controllability enforcement and representational flexibility.*
>
> **Answer:** Thank you. We kindly refer the reviewer to our response to the earlier comment for a detailed explanation.
>
> ---
> **Question:** *The current method enforces controllability but not stability. Have you observed instability in the learned latent dynamics (e.g., exploding eigenvalues of $A_θ$)? It would be useful to regularize eigenvalues of $A_θ$ or provide evidence that the MPC formulation mitigates instability during rollout.*
>
> **Answer:** Thank you. We kindly refer the reviewer to our response to the earlier comment. Our method imposes no constraints on the eigenvalues of $A_θ$. Closed-loop stability is ensured by the MPC controller used during control design.
>
> ---
>
> **Question:** *The Gramian term involves computing $W_T^y$. How is this computed efficiently during training, and how does it scale with the latent dimension $N$? It would be helpful to include complexity analysis or ablation to show that the regularizer does not dominate training time.*
>
> **Answer:** Thank you for your question, we kindly refer the reviewer to our response to the earlier comment for a detailed explanation.

---

> ### Author Response · Authors · 2025-11-25
>
> ---
>
> **Question:** *The framework is called “Koopman,” but it remains unclear whether the learned observables correspond to Koopman-invariant subspaces or simply linear latent embeddings. Can you empirically or theoretically justify that $φ_θ(x)$ approximates Koopman eigenfunctions or invariant coordinates?*
>
> **Answer:** The theoretical and conceptual justification is discussed above in our response on Koopman invariance. Empirically, we also compare our method to EDMD with widely used RBF dictionaries as in [1]:
>
> **Table 1: Evaluation Results (15th step prediction error)**
>
> |         | Mountain Car | Pendulum | Cartpole |
> |---------|--------------|----------|----------|
> | RBF-EDMD | 0.1233      | 2.624    | 0.0285   |
> | Ours    | 0.000117     | 0.0326   | 0.00029  |
>
> This comparison highlights that our learned representation can yield more accurate predictive performance than standard EDMD with RBF dictionaries, even though RBF dictionaries are strong approximators.
>
> [1] Korda, M. and Mezić, I., 2018. Linear predictors for nonlinear dynamical systems: Koopman operator meets model predictive control. Automatica, 93, pp.149-160.
>
> ---
>
>
>
> **Question:** *Can your framework scale to higher-dimensional or partially observed systems (e.g., PDEs, power network, or soft-robot system)? Discuss computational limits and whether encoder-decoder architectures can handle such cases without losing controllability guarantees.*
>
> **Answer:** We appreciate the reviewer’s suggestion. We have show the additional experiments in detail in the previous comment section above. General encoder-decoder architectures without our controllability design cannot guarantee controllability in theory, especially under the case that the data is very limited.
>
> We would also like to recognize that system identification for extremely complex and strongly nonlinear dynamics is itself a very challenging problem especially when it is also needed for downstream control tasks, independent of the choice of representation. Koopman-based approaches add an additional structural constraint: the dynamics must admit a useful linear representation in the lifted space. This structural assumption is precisely what enables efficient LQR/MPC design, but it also limits the expressive power of the model. Addressing this limitation is not specific to our method; it is a fundamental property of the Koopman framework.  If learning is limited to specific tasks with specific policies generated dataset as some work did, it becomes much easier, and our method would also apply.
>
> More expressive neural architectures (e.g., attention mechanisms, temporal feature models, sequence encoders) may help improve the quality of the lifting map, but exploring such architectures is orthogonal to our contribution. Our framework is compatible with any such improvements: as long as the learned model retains the core Koopman structure (a linear dynamics in the lifted coordinates), our framework can be applied. Investigating richer lifting architectures is an interesting future direction, but beyond the scope of the present work.
>
> ---
> ---
>
> **Question:** *The paper does not compare against recent Koopman or geometric control approaches (e.g., KEEC) that emphasize structural invariance. What are the pros and cons of controllability-based representations and equivariance-based representations?*
>
> **Answer:** Thank you for bringing these references to our attention. KEEC considers properties such as equivariance and isometry, and provides a powerful way to incorporate these structures into Koopman learning. The controllability based representation we propose does not conflict with equivariance based representations. Similar to KEEC, our work enforces equivariance through the prediction loss:
>
> $$
> L_{pred}(\theta) =
> \frac{1}{t_f - t_0}
> \int_{t_0}^{t_f}
> w(t)\
> \|\hat{x}(t) - x(t)\|_2^2
> \, dt
> $$
>
>
> where $\hat{x}(t) = C z(t)$ with initial state $z(t_0) = Ψ_θ(x(t_0))$.
>
> In KEEC, equivariance is achieved through the loss:
>
> $$
> L_{equiv}(\phi)=\sum_{t=t_0}^{t_0 + (L-1)\delta t }\|g_{\Phi}^{de}(\hat z_{t+\delta t}) - x_{t+\delta t} \|_2^2
> $$
>
>
> where $\hat z_{t+\Delta t}$ is obtained from the zeroth order hold (ZOH) method from the flow of $z(t) = g_\phi^{\mathrm{en}}(x(t))$. Here, $g_\phi^{\mathrm{de}}(\cdot)$ is the decoder and $g_\phi^{\mathrm{en}}(\cdot)$ is the encoder. In the sense of ensuring equivariance, the main differences between our work and KEEC are the following: i) we consider a neural ODE formulation, while KEEC considers ZOH, and ii) we ensure identity lift through a linear decoder $C \cdot$, while KEEC implements this using a standard identity loss $|g_\phi^{\mathrm{de}} \circ g_\phi^{\mathrm{en}}(x_t) - x_t|$ in an auto encoder. The controllability is ensured through the structure of $A_\theta$ and $B_\theta$.

---

> ### Author Response · Authors · 2025-11-25
>
> For the impact of the hard controllability based representations on the soft equivariance based representations, the following qualitative statements can be made. First, when the amount of training data is sufficiently large and the dimension of the latent space is also sufficiently large, enforcing the equivariance loss alone yields a high precision approximation. Second, when the amount of data is not sufficiently large, incorporating the controllability structure restricts the search to controllable models, thereby greatly reducing the parameter space. This improves data efficiency in terms of equivariance precision. We discussed this in Figure 1 of the introduction section and observed both statements in our simulations.
>
> Other structural priors, such as isometry, which is integrated by KEEC, would be an interesting extension of our work. We thank the reviewer again for providing these references, and we will include a discussion of KEEC in the manuscript.
>
> ---
>
> ---
>
> **Question:** *Parameters like the Gramian regularization weight and time horizon T may strongly affect results. How sensitive is performance to these settings? Suggest including an ablation or at least a qualitative discussion on how these hyperparameters influence both controllability conditioning and prediction accuracy.*
>
> **Answer:** We thank the reviewer for raising this point. We clarify how the finite horizon $T$ and the Gramian regularization weight $\lambda_{\text{gram}}$ influence our method.
>
> **The horizon $T$**:
> It specifies the finite time window over which the controllability Gramian is evaluated. As in standard finite-horizon controllability analysis, increasing $T$ incorporates the dynamics over a longer interval and correspondingly emphasizes the controllability structure more strongly. This produces a predictable and monotonic effect on the Gramian that larger $T$ captures more of the system’s evolution and strengthens the regularization, which is related though not equivalent, to increasing the regularization weight $\lambda_{\text{gram}}$.
>
> In practice, we set $T$ to match the prediction horizon used by the Koopman model and by MPC, so that the controllability shaping is applied over the same time scale on which the model is actually utilized for control.
>
> **Regularization weight $\lambda_{gram}$**:
> We performed experiments with different values of $\lambda_{\text{gram}}$ in our original implementation, and we have now moved these results with figures and analysis to the Appendix G for clarity. As the weight increases, the optimization places more emphasis on improving controllability metrics. However, this comes with the expected trade-off: a stronger controllability bias slightly increases the prediction error, and the MPC controller then compensates for this with correspondingly larger control inputs. Therefore, the final closed-loop behavior is determined by a balance between prediction accuracy and controllability shaping, rather than by either factor alone.
>
> ---
>
>
> **Question:** *The paper claims the learned model can be “directly integrated” with MPC. Was this tested in a closed-loop simulation with constraints? Suggest presenting control performance metrics (tracking error, energy use, robustness) under realistic MPC settings to demonstrate deployability.*
>
> **Answer:** Thank you. We have already included energy usage and target tracking performance in our paper, as shown in Table 2 and Figure 7. As to constraints, the Koopman MPC formulation in [1] explicitly supports state and input constraints, and therefore constrained MPC is naturally admissible in our setting as well especially our lifted state contains the identity lift of original state (i.e., $x = C z$ with $C = [I &ensp;  0]$). If needed, more general nonlinear constraint functions could also be included as additional observables in the lifting map, following the construction used in their paper.
>
> [1] Korda, Milan, and Igor Mezić. "Linear predictors for nonlinear dynamical systems: Koopman operator meets model predictive control." Automatica 93 (2018): 149-160.
>
> ---
>
> We are grateful for the reviewer's very valuable feedback. We hope our clarifications help address the concerns raised.

---

### Official Review · Reviewer_vF1q · 2025-11-03

**Soundness:** 4
**Presentation:** 4
**Contribution:** 3
**Rating:** 8
**Confidence:** 3

**Summary:**

This paper presents a dynamics learning method that enforces controllability as a structural prior in order to improve the quality and usability of the learned dynamics. In particular, this paper:
* Introduces a Koopman-based framework for learning linear surrogate models using Neural ODEs.
* Enforces controllability of the learned model by design, by proposing a particular parameterization of the to-be-learned Koopman operator. This parameterization is based on the observation that enforcing state-to-output controllability in the lifted space (easy) is equivalent to enforcing output-to-output controllability in the original space (hard).
* Provides experiments on four control settings (mountain car, pendulum, cartpole, gene regulatory network), showing improved quality of the learned dynamics and improved controllability of those dynamics compared to SOTA baselines.

**Strengths:**

* The idea of the paper is clear, simple, and well-grounded: Encode controllability as a structural prior enforced by design within neural ODE learning. The method and motivation are very clearly explained, and the clear notation facilitates understanding of how the Koopman operator is integrated into the neural ODE framework.
* The proposed parameterization is backed by theoretical proofs, proving controllability.
* The proposed loss regularization is a nice idea to ensure not just the binary notion of "controllable," but also improve the degree of controllability.
* The experimental results on mountain car, pendulum, and cartpole compare against SOTA baselines, and clearly show improved quality of dynamics learning in low-data regimes (comparable quality in higher-data regimes), as well as improve controllability of the learned dynamics when used within MPC.

**Weaknesses:**

* The experimental settings are all quite small, with the largest setting (GRN = gene regulatory network) including 6 dimensions and 3 control inputs, and with the best results on the smaller single-input settings (GRN is the only multi-input setting, and results are more more marginal). It would be nice to see additional and more convincing validations on larger systems.
* For large systems, the eigenvalue computation needed for the Gramian regularization term might be expensive to compute. In general, it would be nice to see relative training cost between the different methods reported.

**Questions:**

* What do results look like on larger experimental systems?

---

> ### Author Response · Authors · 2025-11-25
> **Response to Reviewer vF1q**
>
> ### Summary
>
> We thank the reviewer for their acknowledgment of our work. Following the reviewer’s comments, we have added additional experiments on larger nonlinear systems, including the Reacher environment in Mujoco and a 7 DoF robotic manipulator. We have also included an ablation study to report the training cost.
>
> ---
>
> ### Response to Weaknesses
>
> **Comment:** *The experimental settings are all quite small, with the largest setting (GRN = gene regulatory network) including 6 dimensions and 3 control inputs, and with the best results on the smaller single-input settings (GRN is the only multi-input setting, and results are more more marginal). It would be nice to see additional and more convincing validations on larger systems.*
>
> **Response:** We appreciate the reviewer’s suggestion. Our experimental evaluation includes standard control benchmarks widely used in the Koopman literature (Mountain Car, Inverted Pendulum, and CartPole), as well as the gene regulatory network (GRN) system, which is a less commonly explored domain in Koopman-based work. For context, the recent ICLR 2025 paper [1], as cited in our manuscript, is one of the few works that also includes the GRN system and similarly evaluates on these benchmark tasks without extending to more complex robotic systems. To our knowledge, experiments on highly complex robotic systems where data generated by random inputs are rare in the current Koopman literature. Given this context, we believe our evaluation provides a fair comparison on the benchmarks that are established and widely used in the field.
>
> We understand that the reviewer has higher expectations for our method, and we share the enthusiasm about testing our framework on more complex systems. To address this, we have conducted additional experiments on two more complex robotic systems: Reacher from the MuJoCo suite and the Franka robot with 7 degrees of freedom [2].
>
> A summary of the new results is:
> **Table: Comparison of prediction error across environments with varying fractions of training data.**
>
> | Environment | Method | 1%             | 5%       | 10%      | 30%      | 50%                 | 100%                |
> |-------------|--------|----------------|----------|----------|----------|----------------------|----------------------|
> | **Reacher** | DKO    | >5             | 3.47     | 1.55     | 0.5613   | 0.2572               | 0.0194               |
> |             | Ours   | >5             | **0.0199**   | **0.0135**   | **0.0039**   | **≤1×10⁻⁴**              | **≤1×10⁻⁴**              |
> | **Franka**  | DKO    | 0.0453         | 0.022    | 0.0068   | 1×10⁻⁴   | ≤1×10⁻⁵              | ≤1×10⁻⁵              |
> |             | Ours   | **0.039**          | **0.00064**  | **≤1×10⁻⁵**  | **≤1×10⁻⁵**  | **≤1×10⁻⁵**              | **≤1×10⁻⁵**              |
>
> We have put this table and together with figures on the prediction performance and training loss in Appendix G in out revised manuscript. These results show that our controllability structure remain effective as system complexity increases. Together with our existing evaluation across mechanical control tasks and biological systems, this provides a comprehensive assessment of our framework’s applicability within the scope where Koopman operator methods are currently viable.
>
> [1] Li, Zhaoyang, Minghao Han, and Xunyuan Yin. "MamKO: Mamba-based koopman operator for modeling and predictive control." The Thirteenth International Conference on Learning Representations. 2025.
>
> [2] Shi, Haojie, and Max Q-H. Meng. "Deep Koopman operator with control for nonlinear systems." IEEE Robotics and Automation Letters 7.3 (2022): 7700-7707.
>
> ---

---

> ### Author Response · Authors · 2025-11-25
>
> ---
> **Comment:** *For large systems, the eigenvalue computation needed for the Gramian regularization term might be expensive to compute. In general, it would be nice to see relative training cost between the different methods reported.*
>
> **Response:** We thank the reviewer for this good question. There are many ways to avoid explicitly computing matrix exponentials. While the Gramian $$W_c(T) = \int_0^T e^{At} BB^\top e^{A^\top t} dt$$
> can be written formally in terms of matrix exponentials, there exist multiple standard methods to compute it without ever evaluating $e^{At}$.
>
> Our implementation uses a *coupled ODE system*: we evolve the state transition matrix
> $$\dot{\Phi}(t) = A\Phi(t), \quad \Phi(0) = I,$$
> and simultaneously integrate
> $$\dot{Q}(t) = \Phi(t) BB^\top \Phi(t)^\top, \quad Q(0) = 0,$$
> so that $W_c(T) = Q(T)$. Alternatively, one can directly integrate the Lyapunov differential equation
> $$\dot{W}_c(t) = AW_c(t) + W_c(t)A^\top + BB^\top, \quad W_c(0) = 0.$$
>
> Both approaches require only standard matrix multiplications at each integration step, avoiding any matrix exponential computations.
>
> **Empirical overhead is modest.**
> Using the pendulum experiment ($n_z = 18$) as an example for ablation, and with our problem size and batch size (64), this step contributes only a modest fraction of the overall cost. Empirically, enabling the Gramian regularizer increases wall clock training time by a factor of approximately from 1.2 to 1.3 times compared to training without it (see the table for detailed timings across experiments). The table also shows that training remains fast even when the Gramian regularizer is included.
>
> **Table: Training time w/ and w/o computation of Gramian**
>
> | Experiment | Gramian | 10% | 30% | 50% |
> |-----------|---------|-----|-----|-----|
> | Pendulum  | w/o     | 7s  | 23s | 38s |
> |           | w/      | 9s  | 29s | 48s |
>
> **Overall training time remains practical.**
> As shown in the table, training remains fast even when the Gramian term is used. Crucially, our framework achieves good predictive accuracy with fewer data samples and fewer training iterations compared to unconstrained approaches, as shown in Table 1 of our manuscript. This sample efficiency, enabled by the control theoretic constraints, means that despite the approximately 1.2 times overhead, the total wall clock time required to reach a given level of performance remains competitive or even favorable. The modest computational cost of the Gramian term is therefore offset by faster convergence and reduced data requirements.
>
> **Further acceleration is possible.**
> If desired, the Gramian can be computed without any numerical integration using Van Loan’s block matrix exponential formula [1]. Defining
> $$
> M =
> \begin{bmatrix}
> A & BB^\top \\\\
> 0 & -A^\top
> \end{bmatrix}
> $$
>
> $$
> e^{MT} =
> \begin{bmatrix}
> \Phi_{11}(T) & \Phi_{12}(T) \\\\
> 0            & \Phi_{22}(T)
> \end{bmatrix}
> $$
> we have $W_c(T) = \Phi_{12}(T) \Phi_{11}(T)^\top$, which requires only a single $(2 n_z) × (2 n_z)$ matrix exponential per update rather than iterative integration.
>
> [1] Van Loan, Charles. "Computing integrals involving the matrix exponential." IEEE Transactions on Automatic Control 23.3 (1978): 395–404.
>
> ---

---

> ### Author Response · Authors · 2025-11-25
>
> ---
>
> ### Answers to Questions
>
> **Question:** *What do results look like on larger experimental systems?*
>
> **Answer:** Thank you for your question, and we have addressed it in the previous section. We would also like to emphasize that system identification for extremely complex and strongly nonlinear dynamics is inherently a very challenging problem, especially when the learned model must support downstream control tasks. This difficulty arises independent of the choice of representation. Koopman-based approaches introduce an additional structural constraint: the dynamics must admit a useful linear representation in the lifted space. This structural assumption is what enables efficient LQR and MPC design, but it also limits the expressive capacity of the model. Addressing this limitation is not specific to our method; it is a fundamental aspect of the Koopman framework. If learning is limited to specific tasks with specific policies generated dataset as some work did, it becomes much easier, and our method would also apply.
>
> More expressive neural architectures, such as attention mechanisms, temporal feature models, or sequence encoders, may help improve the quality of the lifting map. However, exploring such architectures is orthogonal to our contribution. Our framework is compatible with any such improvements. As long as the learned model preserves the core Koopman structure, namely linear dynamics in the lifted coordinates, our approach remains applicable. Investigating richer lifting architectures is an interesting future direction, but is beyond the scope of the present work.
>
> ---
> We sincerely thank the reviewer's time, constructive feedback, and positive assessment. We hope our clarifications help address the concerns raised.

---

> > ### Comment · Reviewer_vF1q · 2025-11-27
> >
> > Thank you to the authors for their response. I maintain my positive impression of the paper.
> >
> > Note: I am not an expert in control or Koopman methods -- the points other reviewers raise seem to be largely about the relationship of this work to the broader Koopman methods literature, which I cannot substantively comment on. My review comes from the perspective of the deep learning-with-constraints literature, and from that perspective, I think this paper provides a solid contribution that nicely integrates control-based results into the design of deep learning architectures.

---

> > > ### Author Response · Authors · 2025-11-27
> > >
> > > Dear reviewer,
> > >
> > > We want to thank you again for your prompt response and positive impression for our work. Your summary is exact: we want to bring domain knowledge from control to incorporate hard constraints into deep learning frameworks.
> > >
> > > All the best,
> > > The Authors.

---

### Official Review · Reviewer_x559 · 2025-11-05

**Soundness:** 3
**Presentation:** 3
**Contribution:** 1
**Rating:** 4
**Confidence:** 5

**Summary:**

The paper consider the problem of modeling and control of nonlinear systems via Koopman operator theory (KOT). Particularly, the paper ensures controllability by using a certain canonical parameterization and also are able to tune the degree of controllability to ensure better control performance. In addition they use Neural ODE to learn a KOT based model Simulation results present the efficacy of their approach

**Strengths:**

The main strengths are as follows:
1) The efficacy of their approach over other methods is clearly show.
2) The literature review in the introduction is extensive.
3) The use of Neural ODE is a smart choice over MLPs; one main reason being free from what discretization issues while converting the continuous to discrete systems.

**Weaknesses:**

I have worked on Koopman operators for a while now. I believe these are the following weakness:

1) The KOT for control literature is quite rich today. For instance works that consider controllability for KOT based models, use Neural ODE or similar techniques exist in literature.
2) The motivation for this work is not clear to me.
3) The approximation error between the KOT and real system is not considered. This is particularly important because Koopman operator is an infinite dimensional linear operator and to make it of practical value, a finite approximation of the Koopman operator via EDMD is usually made.
4) Missing real world examples of more sophisticated nonlinear systems such as quadruped or humanoid is missing. There are already hundreds of paper that consider standard and simple nonlinear systems.

**Questions:**

I have the following questions:

1) Since the Koopman model is just a surrogate model, there is an approximation error between actual and approximated model. This is not considered in the paper. In addition, there exists works that consider this approximation rigorously (see [1]).
2) There are also papers that consider the controllability of the KOT surrogate model. It is not clear to me on how your work is significantly different from [2].
3) It is not always possible to make a nonlinear system controllable by ensuring controllability in the KOT model.
4) There has been lots of work that convert unknown dynamics into Koopman based models and use the surrogate model for control design purposes (see [3,4,5] ). It is not clear how your work is different from them. Yes, you mention that you also ensure controllability of the KOT model, but [2] does this as well
5) In simulation results, although you have compared your approach with some prior methods, the nonlinear systems considered are too simple. I would like examples of more complex robotic systems such as quadruped and humanoid where efficiency of KOT models is yet to shown.
6) Showing that the proposed approach works on more complex nonlinear dynamics such as quadruped or humanoid needs to shown. In addition, they must be well motivated to show their efficacy compared to RL based control policies
7) Ensuring controllability as one of your main contributions would not add significant contribution to the paper given the fact that there exists some work that do it



[1] Mamakoukas, G., Di Cairano, S. and Vinod, A.P., 2022, June. Robust model predictive control with data-driven Koopman operators. In 2022 American Control Conference (ACC) (pp. 3885-3892). IEEE.

[2] Choi, Joonwon, Minhyun Cho, Hyunsang Park, Vishnu Vijay, and Inseok Hwang. "On The Controllability Preservation of Koopman Bilinear Surrogate Model." In 2024 IEEE 63rd Conference on Decision and Control (CDC), pp. 3457-3462. IEEE, 2024.

[3] Korda, M. and Mezić, I., 2018. Linear predictors for nonlinear dynamical systems: Koopman operator meets model predictive control. Automatica, 93, pp.149-160.

[4] Zinage, V. and Bakolas, E., 2023. Neural koopman lyapunov control. Neurocomputing, 527, pp.174-183.

[5] Salzmann, T., Kaufmann, E., Arrizabalaga, J., Pavone, M., Scaramuzza, D. and Ryll, M., 2023. Real-time neural mpc: Deep learning model predictive control for quadrotors and agile robotic platforms. IEEE Robotics and Automation Letters, 8(4), pp.2397-2404.

---

> ### Author Response · Authors · 2025-11-25
> **Response to Reviewer x559**
>
> ### Summary
>
> We thank the reviewer for the positive assessment and helpful references. Our work focuses on learning Koopman models that are controllable by design, which provides explicit algebraic guarantees of controllability through a new canonical parameterization of the lifted dynamics, unlike prior approaches.
>
> On the theoretical side, we establish the relationships between SOC and OOC for our Koopman framework, derive a verifiable condition for OOC, and introduce a canonical form that enforces OOC for the learned Koopman system. On the practical side, we integrate these results into the learning framework together with a gramian regularizer, yielding substantially improved data efficiency and downstream control performance.
>
> We now respond to the reviewer’s specific concerns and questions in detail.
>
> ### Response to Weaknesses
>
> **Comment:** *The KOT for control literature is quite rich today. For instance works that consider controllability for KOT based models, use Neural ODE or similar techniques exist in literature.*
>
> **Response:** Thank you for your comment. Indeed, there have been many works that (i) consider controllability for KOT models, but most of these focus on theoretical analysis and do not propose computational methods that ensure controllability, and (ii) adopt continuous time learning, although this is not the standard approach in the Koopman literature. Most existing methods employ discrete time models of the form $z(k + 1) = A z(k) + B u(k)$ and primarily focus on one step prediction under a fixed sampling interval. Continuous time Neural ODE based formulations are less common in this context. Our choice of a continuous time Neural ODE backbone is motivated by practical advantages that are useful for control oriented modeling. It naturally accommodates irregular or multi rate sampling data and allows the learned continuous time dynamics to be used at control frequencies that differ from those in the training data without requiring modification or retraining. These are practical benefits and illustrate why this formulation is appealing for modeling dynamical systems.
>
> We will clarify the distinctions between our work and the references provided by the reviewer in our responses to the corresponding Questions Section below.
>
> ---
>
> **Comment:** *The motivation for this work is not clear to me.*
>
> **Response:** Thank you. The motivation for this work is twofold. On the control side, most optimization-based control design paradigms, such as LQR and MPC, require *controllability* (or stabilizability in some cases) to ensure bounded closed-loop infinite-horizon cost. However, existing methods such as EDMD and deep learning approaches provide no guarantees of controllability for the learned Koopman system. This motivates our method, which learns a Koopman system with controllability through a canonical parameterization. In experiments, our method exhibits better performance than EDMD-based and deep Koopman-operator-based methods.
>
> On the learning side, a broader limitation of existing system identification methods is the difficulty of incorporating structural priors of the nominal system. One of the most important priors is *controllability*. As we discuss in the introduction and in Figure 1 of the manuscript, restricting the search space to controllable models significantly reduces the parameter space, thereby improving data efficiency. This intuition is consistent with our simulation results, where our method consistently achieves much higher data efficiency than the baseline methods.
>
> ---
>
> **Comment:** *The approximation error between the KOT and real system is not considered. This is particularly important because Koopman operator is an infinite dimensional linear operator and to make it of practical value, a finite approximation of the Koopman operator via EDMD is usually made.*
>
> **Response:** We agree with the reviewer that incorporating the error between the KOT model and the real system into the control design can be beneficial. Since the focus of this paper is on the learning aspect, and robust control design to address model approximation error is well established in the literature, we plan to integrate this consideration in future work.
>
> ---
>
> **Comment:** *Missing real world examples of more sophisticated nonlinear systems such as quadruped or humanoid is missing. There are already hundreds of paper that consider standard and simple nonlinear systems.*
>
> **Response:** We have followed the reviewer’s suggestion and added more complex experiments in the Mujoco environment which are shown in the Questions Section. However, we would like to kindly clarify that the nonlinear systems considered in our paper are benchmark tasks that are widely used and compared in existing works.
>
> ---

---

> ### Author Response · Authors · 2025-11-25
>
> ### Answers to Questions
>
> **Question:** *Since the Koopman model is just a surrogate model, there is an approximation error between actual and approximated model. This is not considered in the paper. In addition, there exists works that consider this approximation rigorously (see [1]).*
>
> **Answer:** Thank you. As we have explained earlier, this paper focuses on ensuring controllability of the learned Koopman system. This is achieved through a novel canonical parameterization in the latent space together with a controllability enhancement regularizer. The most important feature of our approach is that *regardless of the magnitude of the approximation error, the learned model is always output-to-output controllable*.
>
> We agree with the reviewer that analyzing approximation error can be useful for subsequent robust control design. The approach proposed in [1] can be directly combined with our method. Specifically, the Lipschitz constant of the modeling error can be upper bounded by $L_f + | C A_θ B_θ |$, where $L_f$ is the Lipschitz constant of the nominal dynamics, $C$ is the output matrix, and $A_θ$ and $B_θ$ are the learned matrices obtained by our canonical parameterization.
>
> We plan to conduct experiments using similar robust control schemes in future work, as this direction is valuable but beyond the scope of the present paper.
>
> ---
>
> **Question:** *There are also papers that consider the controllability of the KOT surrogate model. It is not clear to me on how your work is significantly different from [2].*
>
> **Answer:**
> Thank you for pointing out the relevant literature. The two works are fundamentally different. In brief, [2] is built on the assumptions that the ***bilinear*** KOT model is either accurate (Proposition 1 and Corollary 1) or probabilistically accurate (Proposition 2). Under these assumptions, [2] studies *what additional conditions the observables or dictionaries must satisfy so that there ***exists*** a Koopman system is OOC or probabilistically OOC*. However, no numerical methods are proposed to enforce these assumptions during design, nor are practical procedures provided to verify them a posteriori.
>
> In contrast, our work identifies *the relationship between different types of controllability for Koopman systems* and proposes *canonical parameterizations of the learned state and input matrices that provide practical controllability guarantees*.
>
> On the theoretical side, our method does not require the amount of training data to be sufficiently large, which is an assumption made in [2]. This is because our approach does not rely on any accuracy assumptions between the KOT model and the nominal system, making it applicable even when data are limited. On the practical side, our canonical parameterization can be directly integrated into the learning framework, whereas [2] remains purely theoretical. In fact, none of the assumptions proposed in [2] are verified in their experiments. These aspects highlight the significant theoretical and practical differences between our work and [2], as well as the advantages of our approach in terms of applicability.
>
> The following table clearly summarizes the differences, which we believe are substantial.
>
> |               | Controllability | Theory                          | Computation                                       | Experiments |
> |---------------|------------------|----------------------------------|----------------------------------------------------|-------------|
> | Our Work      | OOC/SOC          | Over state and input matrices    | Canonical parameterization and regularizer under **machine learning** framework         | Validated   |
> | [2]           | OOC              | Over dictionaries/observables    | N/A (only theoretical results on ***existence*** under assumptions and no **learning**)   | N/A         |
>
> Another important feature of our method is data efficiency. As stated in the paper and illustrated especially in Figure 1, our structural design improves data efficiency and enables the model to learn a good representation much faster. This aspect is not studied in [2].
>
> To summarize:
>
> - The aims of the two papers are different. The goal of [2] is to study under what conditions OOC can be preserved, whereas our goal is to incorporate OOC during training so that learning becomes more effective and downstream control performance is improved.
> - The fundamental assumptions of the two papers are different. [2] assumes that the KOT model is either accurate, corresponding to a model-based setting, or accurate within certain error bounds, which requires sufficiently large amounts of data. Our work considers a more practical scenario and makes no assumptions regarding learning precision.
> - The contributions of the two papers are different. [2] focuses solely on theoretical analysis, while our work contributes both theoretically and on the application side, particularly in learning.

---

> ### Author Response · Authors · 2025-11-25
>
> We believe this makes clear that the two works address significantly different research questions. Nevertheless, since studies on the controllability of KOT are limited and [2] is closely related, we appreciate the reviewer for pointing it out. We will add a discussion of this paper in the Introduction and Related Work section, which will help clarify our contributions and further improve the paper.
>
> ---
>
> **Question:** *It is not always possible to make a nonlinear system controllable by ensuring controllability in the KOT model.*
>
> **Answer:**
> We appreciate the reviewer’s question and would like to clarify the intended meaning. If by “nonlinear system” the reviewer is referring to the nominal system, we note that the manuscript already assumes it to be locally controllable after Equation (1): *“Specifically, we consider system (1) where $f(x,u)$ is unknown but locally controllable a priori.”* A similar assumption is also made in reference [2] cited by the reviewer. Under this assumption, prior work has shown that the learned KOT model can be OOC. In our framework, OOC of the learned KOT model is guaranteed whenever the matrices $A_\theta$ and $B_\theta$ satisfy the proposed canonical parameterization. We do not “make” a nonlinear system to be controllable.
>
> ---
>
> **Question:** *There has been lots of work that convert unknown dynamics into Koopman based models and use the surrogate model for control design purposes (see [3,4,5] ). It is not clear how your work is different from them. Yes, you mention that you also ensure controllability of the KOT model, but [2] does this as well*
>
>
> **Answer:**
> We appreciate these references. As emphasized in our previous responses, our work focuses on ensuring controllability of the learned nonlinear dynamics by enforcing OOC of the KOT model. The distinction between our work and [2] has already been clearly discussed, and we note again that [2] does not present any computational results due to the theoretical nature of its analysis.
>
> Looking more closely at [3, 4, 5], these papers address topics that differ from ours. *Our work focuses on learning a controllable KOT model*, whereas all of these works focus on *controller design*. Reference [3] is a pioneering contribution that first introduced the Koopman MPC framework. Reference [4] proposes designing stabilizing controllers by learning Lyapunov functions for the KOT model. Neither work considers controllability analysis, controllability preservation by design, or controllability enhancement. Our work addresses these problems through the proposed canonical parameterization and regularizer. Reference [5] does not consider Koopman operator theory at all; instead, it focuses on efficiently integrating neural networks as dynamical models within MPC pipelines. The discussion of [3] has already been in our original manuscript, and we also have included [2][4] in our revised manuscript.
>
> This highlights that our work addresses a distinct problem that is not covered in the referenced papers.
>
> [1] Mamakoukas, G., Di Cairano, S. and Vinod, A.P., 2022, June. Robust model predictive control with data-driven Koopman operators. In 2022 American Control Conference (ACC) (pp. 3885-3892). IEEE.
>
> [2] Choi, Joonwon, Minhyun Cho, Hyunsang Park, Vishnu Vijay, and Inseok Hwang. "On The Controllability Preservation of Koopman Bilinear Surrogate Model." In 2024 IEEE 63rd Conference on Decision and Control (CDC), pp. 3457-3462. IEEE, 2024.
>
> [3] Korda, M. and Mezić, I., 2018. Linear predictors for nonlinear dynamical systems: Koopman operator meets model predictive control. Automatica, 93, pp.149-160.
>
> [4] Zinage, V. and Bakolas, E., 2023. Neural koopman lyapunov control. Neurocomputing, 527, pp.174-183.
>
> [5] Salzmann, T., Kaufmann, E., Arrizabalaga, J., Pavone, M., Scaramuzza, D. and Ryll, M., 2023. Real-time neural mpc: Deep learning model predictive control for quadrotors and agile robotic platforms. IEEE Robotics and Automation Letters, 8(4), pp.2397-2404.
>
> ---
>
> **Question:** *Ensuring controllability as one of your main contributions would not add significant contribution to the paper given the fact that there exists some work that do it.*
>
> **Answer:**
> Thank you for your question. As discussed in our responses to previous comments and questions, our work is the *first* to analyze the connections between SOC and OOC for KOT at the theoretical level, and the ***first*** to propose algebraic conditions and computational methods that rigorously ensure OOC of the KOT model. No other work did this before. The proposed method not only improves data efficiency during learning, but also yields superior control performance when using the learned OOC KOT model. To our knowledge, none of the existing works have conducted similar analyses or proposed comparable computational methods.
>
> ---

---

> ### Author Response · Authors · 2025-11-25
>
> ---
>
> **Question:** *In simulation results, although you have compared your approach with some prior methods, the nonlinear systems considered are too simple. I would like examples of more complex robotic systems such as quadruped and humanoid where efficiency of KOT models is yet to shown. In addition, they must be well motivated to show their efficacy compared to RL based control policies.*
>
> **Answer:**
> We appreciate the reviewer’s suggestion. Our experimental evaluation includes standard control benchmarks widely used in the Koopman literature (Mountain Car, Inverted Pendulum, and CartPole), as well as the gene regulatory network (GRN) system, which is a less commonly explored domain in Koopman based research. For context, the recent ICLR 2025 paper [1], cited in our manuscript, is one of the few works that also includes the GRN system and likewise evaluates on these benchmark tasks without extending to quadruped or humanoid robots. To our knowledge, as the reviewer correctly pointed out, experiments on such high complexity robotic systems, including humanoids, where data generated by random inputs are rare in the current Koopman literature. Given this context, we believe our evaluation provides a fair comparison on the benchmarks that are established and widely used in the field.
>
> We understand that the reviewer has higher expectations for our method, and we share the enthusiasm for evaluating our framework on more complex systems. To address this, we have conducted additional experiments on two higher complexity robotic systems: Reacher from the MuJoCo suite and the Franka robot with 7 degrees of freedom [2].
>
> **Table: Comparison of prediction error across environments with varying fractions of training data.**
>
> | Environment | Method | 1%             | 5%       | 10%      | 30%      | 50%                 | 100%                |
> |-------------|--------|----------------|----------|----------|----------|----------------------|----------------------|
> | **Reacher** | DKO    | >5             | 3.47     | 1.55     | 0.5613   | 0.2572               | 0.0194               |
> |             | Ours   | >5             | **0.0199**   | **0.0135**   | **0.0039**   | **≤1×10⁻⁴**              | **≤1×10⁻⁴**              |
> | **Franka**  | DKO    | 0.0453         | 0.022    | 0.0068   | 1×10⁻⁴   | ≤1×10⁻⁵              | ≤1×10⁻⁵              |
> |             | Ours   | **0.039**          | **0.00064**  | **≤1×10⁻⁵**  | **≤1×10⁻⁵**  | **≤1×10⁻⁵**              | **≤1×10⁻⁵**              |
>
> We have put this table and together with figures on the prediction performance and training loss in Appendix G in out revised manuscript. These results demonstrate that our controllability structure remain effective as system complexity increases. Together with our existing evaluation across mechanical control tasks and biological systems, this provides a comprehensive assessment of the applicability of our framework within the scope where Koopman operator methods are currently viable.
>
> **On comparison with RL methods.**
> Regarding comparison with reinforcement learning approaches, we note that RL methods and Koopman based methods address different problem settings and evaluation criteria. RL methods learn policies through extensive interaction with the environment, typically requiring millions of samples, whereas our approach learns predictive models from limited trajectory data for model based control. Direct comparison is therefore not straightforward, as the two types of methods optimize different objectives under different data assumptions. Our framework is more naturally compared against other model based learning methods, which we have evaluated extensively in the main paper.
>
> We would also like to acknowledge that system identification for extremely complex and strongly nonlinear dynamics is itself a very challenging problem, especially when the learned model must support downstream control tasks. This challenge exists regardless of the choice of representation. Koopman based approaches introduce an additional structural constraint: the dynamics must admit a useful linear representation in the lifted space. This structural assumption is what enables efficient LQR and MPC design, but it also limits the expressive capacity of the model. Addressing this limitation is not specific to our method. If learning is limited to specific tasks with specific policies generated dataset as some work did,  it becomes much easier, and our method would also apply.
>
> More expressive neural architectures, such as attention mechanisms, may help improve the quality of the lifting map. However, exploring such architectures is orthogonal to our contribution. Our framework is compatible with any such improvements as long as the learned model preserves the core Koopman structure, namely linear dynamics in the lifted coordinates. Investigating richer lifting architectures is an interesting direction for future work, but is beyond the scope of the present work.

---

> > ### Author Response · Authors · 2025-11-25
> >
> > [1] Li, Zhaoyang, Minghao Han, and Xunyuan Yin. "MamKO: Mamba-based koopman operator for modeling and predictive control." The Thirteenth International Conference on Learning Representations. 2025.
> >
> > [2] Shi, Haojie, and Max Q-H. Meng. "Deep Koopman operator with control for nonlinear systems." IEEE Robotics and Automation Letters 7.3 (2022): 7700-7707.
> >
> > ---
> > We sincerely thank the reviewer for the comments and suggestions. We hope our responses have fully addressed all concerns and further clarified our contributions.

---

### Official Review · Reviewer_ybHH · 2025-11-08

**Soundness:** 2
**Presentation:** 2
**Contribution:** 1
**Rating:** 2
**Confidence:** 4

**Summary:**

The paper proposes a Koopman-based representation learning framework, implemented as an end-to-end Neural ODE, that learns nonlinear dynamical models from limited data while ensuring the learned models remain suitable for control.  It enforces controllability by construction via a new canonical parameterization of the latent linear dynamics (A, B), and shows that controllability of the learned latent model implies controllability of the original (nominal) system.  The method also shapes the degree of controllability by adding a finite-horizon output-Gramian regularizer that enlarges the smallest eigenvalue and controls the condition number to promote well-conditioned models for control. For downstream control, the learned linear surrogate enables a convex quadratic-program MPC in the lifted space with receding-horizon execution.  Empirically, on standard nonlinear benchmarks (pendulum, mountain car, cartpole), the approach achieves more accurate long-horizon prediction and better MPC performance with improved data efficiency compared to Deep Koopman Operator and MLP baselines.

**Strengths:**

1. Learning the Koopman operator is an interesting and important classical control system problem.

**Weaknesses:**

1.	This paper has marginal algorithmic or theoretical contributions. See the details below.
2.	The main contribution is a method to learn the Koopman operator. However, the proposed methods are quite standard, typical in any Koopman operator learning approach. While the paper prominently states the connection with Neural ODE, it is not really clear why that is relevant or interesting. Lifting the state x to a high-dimensional variable z and modeling the non-linear system as a linear system at this lifted space is indeed the standard approach of Koopman theory. So, the reason for presenting neural ODE as new approach is not clear.
3.	The theoretical results presented are standard results from linear systems theory. In particular, Theorem 1 about controllability and Theorem 2 about reparameterization to a suitable form are standard results. So, it is not clear why these are presented as novel results, and how they contribute to the Koopman operator theory.
4.	Once the Koopman operator is learned, LQR or MPC are the standard ways to design a control policy. There is no novelty in that part.
5.	The experiments are done on three very simple tasks: mountain car, pendulum, cartpole. It is not clear if the proposed methods can scale to even a slightly more difficult settings, say, mujoco environments.

**Questions:**

1. Aren't Theorems 1 and 2 about any linear systems? Any specific connection to Koopman operators?
2. Will the proposed approach work even in slightly highly dimensional non-linear systems, such as simple MuJoCo environments?

---

> ### Author Response · Authors · 2025-11-25
> **Response to Reviewer ybHH**
>
> ### Summary
>
> We thank the reviewer for their thoughtful summary of our work and for the inspiring questions. As a brief overview, our main contribution is to learn *controllable by design* dynamics using Koopman operator theory.
>
> On the theory part, we
>
> - Establish a connection between State to Output Controllability (SOC) and Output to Output Controllability (OOC) of the Koopman linear system in Lemma 1;
> - Propose a necessary and sufficient condition for verifying OOC for the Koopman linear system in Theorem 1;
> - Introduce a new canonical form as a parameterization that ensures OOC for the learned Koopman linear system in Theorem 2.
>
> We appreciate the reviewer’s concern that these results may appear standard at first glance due to their subtle nature. However, as we clarify in the sequel, none of these results have appeared in the existing literature. More importantly, they provide a ready to integrate and efficient approach for learning nonlinear dynamics with controllability guarantees, which in turn strengthens downstream control design.
>
> Utilizing this theory, we
>
> - Propose an algorithm for learning controllable dynamics by training neural ODEs based on the Koopman representation;
> - Introduce a regularizer that enhances the degree of controllability of the learned dynamics.
>
> The proposed method demonstrates **higher** data efficiency, measured by the amount of data required to learn the dynamics to a given precision, and **better** control performance in terms of MPC cost, compared with existing work in the deep Koopman operator literature.
>
> Following the reviewer’s comments, we have also conducted additional experiments on Mojoco. In these new experiments, we again observe improved data efficiency and better control performance relative to baseline deep Koopman approaches.
>
>
> ---
>
> ### Response to Weaknesses
>
> **Comment:** *This paper has marginal algorithmic or theoretical contributions. See the details below.*
>
> **Response:** We will address each of the reviewer's concerns in detail.
>
> ---
>
> **Comment:** *The main contribution is a method to learn the Koopman operator. However, the proposed methods are quite standard, typical in any Koopman operator learning approach. While the paper prominently states the connection with Neural ODE, it is not really clear why that is relevant or interesting. Lifting the state $x$ to a high-dimensional variable $z$ and modeling the non-linear system as a linear system at this lifted space is indeed the standard approach of Koopman theory. So, the reason for presenting neural ODE as new approach is not clear.*
>
> **Response:** As clarified in the summary, our objective is to learn nonlinear dynamics with controllability guarantees. As the reviewer correctly noted, the fundamental techniques we employ are (i) Koopman operator theory, which transforms nonlinear dynamics into linear ones, and (ii) Neural ODE methods, which efficiently learn ODEs. However, a direct combination of these techniques does not yield dynamics that satisfy controllability, which is an essential property for downstream control tasks. Motivated by this gap, our key contribution is a new canonical parameterization for the learned dynamics that ensures *controllability by design*. Building on this parameterization, we establish connections between SOC and OOC and introduce a regularizer that further enhances the degree of controllability during training.
>
> Regarding the appealing feature of Neural ODE for deep Koopman learning, which we believe is new, we would also like to clarify that:
>
> 1. Although we do not claim that Neural ODEs themselves constitute a new approach, we note that continuous time learning is not the standard practice in the Koopman literature. Most existing methods rely on discrete time models of the form $z(k + 1) = A z(k) + B u(k)$ and primarily focus on one step prediction under a fixed sampling interval. Continuous time formulations based on Neural ODEs are less common in this context.
>
> 2. Our selection of a continuous time Neural ODE backbone is motivated by practical advantages that are useful in control oriented modeling. It naturally accommodates irregular or multi rate sampling data and allows the learned continuous time dynamics to be used at control frequencies that differ from those in the training data without requiring modification or retraining. These are practical benefits and the reasons why it is interesting when modeling dynamics.
>
> We hypothesize that the reviewer’s impression that we are presenting Neural ODEs as a new approach may stem from the wording in the first summary point. We will revise the phrasing to make our intention clearer. Thank you again for your comments.
>
> ---

---

> ### Author Response · Authors · 2025-11-25
>
> **Comment:** *The theoretical results presented are standard results from linear systems theory. In particular, Theorem 1 about controllability and Theorem 2 about reparameterization to a suitable form are standard results. So, it is not clear why these are presented as novel results, and how they contribute to the Koopman operator theory.*
>
> **Response:** Thank you for your comments. We will clarify the novelty and originality of each main result. We would like to emphasize that, to the best of our knowledge, ***both Theorem 1 and Theorem 2 are not standard results and are in fact novel contributions***.
>
> The first point we would like to clarify is the distinction between State to Output Controllability (SOC) and Output to Output Controllability (OOC). ***These are two different notions and properties***. The commonly used and well studied concept of Output controllability in control theory corresponds to SOC, whereas OOC has not been extensively investigated. The first, and to our knowledge the only, work that formally introduced OOC and explained its relationship to SOC for general linear systems is [1], published in 2023. We understand this may be surprising. We experienced the same surprise upon discovering that this topic had only very recently been studied, and the same reaction emerged in our discussions with several control theorists.
>
> Moreover, in that prior work, OOC is not a property that can be easily verified, let alone enforced. In contrast, our contribution is to study OOC within the special structure of Koopman linear systems. This perspective is new, and we believe it provides an interesting and useful direction for controllable representation learning.
>
> Below, we clarify the originality of our main results.
>
> - **Lemma 1.** This lemma establishes the necessary and sufficient conditions for SOC and OOC in the Koopman linear system. It is important to note that both SOC and OOC are specialized controllability properties that differ from the standard notion of controllability, which concerns state to state reachability. These two properties are rarely distinguished in most Koopman operator papers. The first work that introduces OOC and explains its relationship with SOC for *general* linear systems is [1] (2023). As stated in that paper, and in our manuscript before Lemma 1, SOC is sufficient but not necessary for OOC. However, our Koopman linear system belongs to a *special class of linear systems* characterized by an identity lift from $\mathbb{R}^n$ (the original space) to $\mathcal{Z}$ (the lifted space). By leveraging this property, together with the fact that the original state $x$ can always be recovered from the lifted state $z$, a property that does not hold for general linear systems, we prove that the Koopman linear system is OOC on $\mathbb{R}^n$ if and only if it is SOC on $(\mathcal{Z}, \mathbb{R}^n)$. This result does not appear in [1] or in any subsequent works.
>
> - **Theorem 1.** This theorem provides an algebraic condition for determining OOC of the Koopman linear system. As discussed in [1], verifying OOC for general linear systems is challenging. The only available conditions are complicated rank criteria that involve time integrations, as presented in Theorem 5.1 of [1] for finite time OOC verification. In contrast, we propose a relatively simple condition, which is also known as the Kalman rank condition for SOC. The reason this condition can be used to test OOC is that: (i) SOC on $\mathbb{R}^n$ is equivalent to OOC on $(\mathcal{Z}, \mathbb{R}^n)$, as established in Lemma 1, and (ii) for linear systems, SOC on $(\mathcal{Z}, \mathbb{R}^n)$ is equivalent to SOC on $(\mathbb{R}^N, \mathbb{R}^n)$. These perspectives have not been presented in the existing literature.
>
> - **Theorem 2.** This theorem introduces a ***new canonical form*** that ensures OOC for the learned Koopman linear system. Since the output $y$ of the Koopman linear system recovers the original state $x$, OOC implies state to state controllability on the $x$ space. The proposed **OOC canonical form** is fundamentally different from the classical state to state controllability canonical form commonly found in textbooks, and to our knowledge it has not been explored in the existing literature. In fact, the proof of OOC under this canonical form relies directly on Theorem 1.
>
> Figure in Appendix C illustrates how the different notions of controllability are connected within our framework. To clarify these relationships and better highlight our theoretical contributions, we will move this figure into the main body of the paper. ***These results are new and can contribute to both control theory and Koopman learning.***

---

> ### Author Response · Authors · 2025-11-25
>
> Regarding our contribution to the Koopman operator theory, our work provides a systematic method for learning a controllable Koopman linear system. This problem has attracted significant interest within the Koopman operator community. For example, the pioneering work [2] studied state to state controllability of Koopman systems and showed that this form of controllability is difficult to preserve. The authors of [3] proposed an infinite dimensional extension of the Lie algebra rank condition to verify controllability, but this condition is difficult to check and even more challenging to enforce by design. More recently, the authors of [4] introduced sufficient conditions based on *exact representation*,  surjectiveness of observables and invariance of dictionaries. However, it remains unclear how to guarantee these conditions by design, and they were not validated in experiments. Our work builds on the notions of OOC and SOC introduced in [1], establishes the relationships between SOC and OOC for the Koopman system, and develops constructive methods that ensure OOC for the Koopman linear system.
>
> To clarify our contribution, we have added the above discussion to the revised manuscript.
>
> [1] Danhane, Baparou, Jérôme Lohéac, and Marc Jungers. "Characterizations of output controllability for LTI systems." Automatica 154 (2023): 111104.
>
> [2] D. Goswami and D. A. Paley, "Bilinearization, reachability, and optimal control of control-affine nonlinear systems: A Koopman spectral approach," IEEE Transactions on Automatic Control, vol. 67, no. 6, pp. 2715–2728, 2021.
>
> [3] W. Zhang and J.-S. Li, "Koopman bilinearization of nonlinear control systems," arXiv preprint arXiv:2211.07112, 2022.
>
> [4] Choi, Joonwon, Minhyun Cho, Hyunsang Park, Vishnu Vijay, and Inseok Hwang. "On The Controllability Preservation of Koopman Bilinear Surrogate Model." In 2024 IEEE 63rd Conference on Decision and Control (CDC), pp. 3457–3462. IEEE, 2024.
>
> ---
>
> **Comment:** *Once the Koopman operator is learned, LQR or MPC are the standard ways to design a control policy. There is no novelty in that part.*
>
> **Response:** We appreciate the comment. We would like to kindly clarify again that our contributions focus on the learning of Koopman operators, while the control implementation relies on standard classical methods such as LQR and MPC, consistent with most other deep Koopman works. ***Because this is exactly why the community is interested in Koopman:  to enable classical LQR/MPC designed for linear systems to be applied effectively to nonlinear systems***. Although many Koopman based control studies adopt similar control paradigms, the lack of controllability guarantees in the learned systems significantly limits their achievable control performance. This limitation is well known in control theory: without controllability, both LQR and MPC can produce unbounded closed loop infinite horizon cost. From this perspective, our method provides a practically valuable system identification approach that supplies downstream control with learned models that satisfy the necessary controllability properties.
>
> ---
> **Comment:** *The experiments are done on three very simple tasks: mountain car, pendulum, cartpole. It is not clear if the proposed methods can scale to even a slightly more difficult settings, say, mujoco environments.*
>
> **Response:** Thank you for the comment. Following the reviewer’s suggestion, we have conducted experiments on more complex cases. Please see our detailed response to the Questions section below.
>
> ---
> ## Answers to Questions
>
> **Question:** *Aren't Theorems 1 and 2 about any linear systems? Any specific connection to Koopman operators?*
>
> **Answer:** These question has been addressed in our previous responses. The quick answer to the first question is No. The first point that needs clarification is that SOC and OOC are two distinct notions and properties. We believe the reviewer’s impression that Theorem 1 is not novel and may hold for any linear system likely arises from a confusion between SOC and OOC. We will make this distinction clearer in the revised manuscript and will move Figure in Appendix into the main body to aid in clarification.
>
> In summary, both theorems build on Lemma 1, which concerns linear systems for which *every output* $y \in \mathbb{R}^n$ *corresponds to one state* $z \in \mathcal{Z}$ *on some well defined set* $\mathcal{Z} \subseteq \mathbb{R}^N$. This property does not hold for arbitrary linear systems, and this is one of the reasons why characterizing OOC for general linear systems is challenging. For Koopman linear systems, however, we can prove that the map $\mathcal{T} : \mathcal{Z} \to \mathbb{R}^n$ defined by $y = \mathcal{T} z$ is bijective. We kindly refer the reviewer to the proof of Lemma 1 in Appendix D of the manuscript.
>
> ---

---

> ### Author Response · Authors · 2025-11-25
>
> ---
>
> **Question:** *Will the proposed approach work even in slightly highly dimensional nonlinear systems, such as simple MuJoCo environments?*
>
> **Answer:** We appreciate the reviewer’s suggestion. Our experimental evaluation includes standard control benchmarks widely used in the Koopman literature (Mountain Car, Inverted Pendulum, and CartPole), as well as the gene regulatory network (GRN) system, which is a less commonly explored domain in Koopman based work. For context, the recent ICLR 2025 paper [1], cited in our manuscript, is one of the few studies that also includes the GRN system, and it likewise evaluates on these benchmark tasks without extending to more complex systems. To our knowledge, experiments on highly complex robotic systems where data generated by random inputs are rare in the current Koopman literature. Given this context, we believe our evaluation provides a fair and meaningful comparison on benchmarks that are well established and widely accepted in the field.
>
> We understand that the reviewer has higher expectations for our method, and we share the enthusiasm for evaluating our framework on more complex systems. To address this, we have conducted additional experiments on two higher complexity robotic systems: Reacher from the MuJoCo suite and the Franka robot with 7 degrees of freedom [2].
>
> The results are summarized below (prediction error vs. fraction of training data):
>
> **Table: Comparison of prediction error across environments with varying fractions of training data.**
>
> | Environment | Method | 1%             | 5%       | 10%      | 30%      | 50%                 | 100%                |
> |-------------|--------|----------------|----------|----------|----------|----------------------|----------------------|
> | **Reacher** | DKO    | >5             | 3.47     | 1.55     | 0.5613   | 0.2572               | 0.0194               |
> |             | Ours   | >5             | **0.0199**   | **0.0135**   | **0.0039**  | **≤1×10⁻⁴**              | **≤1×10⁻⁴**              |
> | **Franka**  | DKO    | 0.0453         | 0.022    | 0.0068   | 1×10⁻⁴   | ≤1×10⁻⁵              | ≤1×10⁻⁵              |
> |             | Ours   | **0.039**          | **0.00064**  | **≤1×10⁻⁵**  | **≤1×10⁻⁵**  | **≤1×10⁻⁵**              | **≤1×10⁻⁵**             |
>
> We have put this table and together with figures on the prediction performance and training loss in Appendix G in out revised manuscript. These results show that our controllability based structure remains effective as system complexity increases. Together with our existing evaluation across mechanical control tasks and biological systems, this provides a comprehensive assessment of the applicability of our framework within the scope where Koopman operator methods are currently practical.
>
> We would also like to acknowledge that system identification for extremely complex and strongly nonlinear dynamics is itself a very challenging problem, especially when the learned model must also support downstream control tasks. This challenge exists regardless of the choice of representation. Koopman based approaches introduce an additional structural constraint: the dynamics must admit a useful linear representation in the lifted space. This structural assumption is what enables efficient LQR and MPC design, but it also limits the expressive capacity of the model. Addressing this limitation is not specific to our method; it is a fundamental characteristic of the Koopman framework.
>
> More expressive neural architectures, such as attention mechanisms, temporal feature models, or sequence encoders, may help improve the quality of the lifting map. However, exploring such architectures is orthogonal to our contribution. Our framework is compatible with any of these improvements. As long as the learned model preserves the essential Koopman structure, namely linear dynamics in the lifted coordinates, our framework remains applicable. Investigating richer lifting architectures is an interesting direction for future work but is beyond the scope of the present study.
>
> [1] Li, Zhaoyang, Minghao Han, and Xunyuan Yin. "MamKO: Mamba-based koopman operator for modeling and predictive control." The Thirteenth International Conference on Learning Representations. 2025.
> [2] Shi, Haojie, and Max Q-H. Meng. "Deep Koopman operator with control for nonlinear systems." IEEE Robotics and Automation Letters 7.3 (2022): 7700-7707.
>
> ---
> We sincerely appreciate the reviewer's insights and hope our clarifications address these concerns, highlighting the rigor and novelty of our contributions.

---

### Author Response · Authors · 2025-11-25
**Summary of Rebuttal**

We sincerely thank all reviewers for their thoughtful comments, constructive suggestions and the positive feedback regarding our theoretical and algorithmic contributions.

During the rebuttal, we focused on clarifying the key ideas of the paper and addressing common concerns raised across the reviews. In particular:
- We clarified the conceptual novelty of our controllability results (SOC/OOC relations, OOC verification, and the proposed canonical form), and their specific connection to Koopman systems for reviewer ybHH and x559. These results are new and can contribute to both control theory and Koopman learning.
- We explained our use of continuous-time Neural ODEs and why this formulation is practically appealing in control-oriented modeling.
- We added new experiments on more complex tasks (Reacher in Mujoco and 7-DoF Franka), showing that our proposed structure continues to offer improved data efficiency and prediction accuracy at higher system complexity. We put the additional results in Appendix G of our revised manuscript.
- We discussed computational cost, clarified how the Gramian is computed efficiently, and provided an ablation study on training overhead as suggested by reviewer dLx9 and vF1q.
- We expanded the discussion of related work, such as controllability and invariance and included the discussion in the Introduction section in our revised manuscript.

We hope that these clarifications and additions results address the reviewers’ concerns and further strengthen the contribution of our work.

---

### Comment · Area_Chair_XQpJ · 2025-11-28
**Please Check the Authors' Responses**

Dear Reviewers,

The authors have posted their responses. Could you please take a moment to review their responses and check whether your concerns have been adequately addressed (if you have not done it yet)? If possible, kindly initiate the discussion at your earliest convenience.

Your timely assistance is essential for keeping the review process on track. Thank you very much for your support and contribution.

Best regards, Your AC

---

### Author Response · Authors · 2025-12-03
**Summary for Area Chair**

Dear Area Chair,

Thank you for taking the time to handle our submission under the unexpected review-reassignment situation. We greatly appreciate your effort. To support your assessment, we provide a concise summary of our contribution, our revisions, and how each reviewer’s concerns have been addressed. A fully revised manuscript has been uploaded accordingly.

---

# Contribution

We present the first learning framework that enforces Output-to-Output
Controllability (OOC) by design in Koopman-based neural ODE models. Koopman OOC corresponds to State-to-State Controllability (SSC) of the original nonlinear system. Our novel OOC canonical form and Gramian regularizer ensure learned models are controllable, enabling reliable classical control methods (LQR/MPC) on nonlinear systems. Our approach reduces search space and achieves superior data efficiency.

---

# Review Response

---

## 1.  Reviewer ybHH (rating 2)

The most important concern the Reviewer ybHH raised is that they claimed our Theorems 1 and 2 are "standard results from linear systems theory" and lack novelty. **This claim is factually incorrect and reflects a misunderstanding of what controllability property we address.**

Also, the reviewer raised questions like "using LQR/MPC is standard" which reveals that they did not appreciate the motivation of extensive research of Koopman itself, and did not appreciate the aim of our paper.

**The core misunderstanding:**
We believe that the reviewer conflated Output-to-Output Controllability (OOC) with standard State-to-State Controllability (SSC). We study Koopman OOC because original system SSC ⟺ Koopman OOC (necessary and sufficient). Studying Koopman SSC would be wrong: it's only sufficient (not necessary) for original system SSC and impractical due to high lifted-space dimensionality. Our Theorems 1 and 2 are both new results. Critically, ***the reviewer did not support their claim that our results are "standard" by providing any reference, precisely because no such references exist.***

Apart from the request for more complex experiments (which we addressed), **every technical concern raised by this reviewer is based on misunderstanding and lacks any supporting evidence.** The reviewer:
- Provided no references to support "standard results" claim
- Did not distinguish between different controllability notions
- Made no technical argument beyond assertion
- Did not engage with our detailed rebuttal before the reverting

**Our response:**

In rebuttal, we ***addressed every question*** the reviewer raised and clarified the concepts, ideas and novelty in our paper.

1. **OOC was only formalized in 2023**. It is recent research [1], not textbook material. Verifying OOC for general linear systems requires complicated criteria involving time integrations. So it cannot be used in our setting.

2. **Our Theorem 1 is the first algebraic OOC verification condition for Koopman systems and it is not for general linear systems.** We leverage the special structure of Koopman systems (including identity lift from original space) to derive a simple condition: in this special Koopman case, OOC coincides with State-to-Output Controllability (SOC). This does ***NOT*** apply to general linear system.

3. **Our Theorem 2 provides the first OOC canonical form for Koopman systems (SOC for linear systems).** Classical canonical forms address SSC (which may be the thing the reviewer thought of); ours addresses OOC for koopman models. These are fundamentally different mathematical objects serving different purposes. And this is the first work that provides this canonical form, cannot be found in any other literature (so that the reviewer could not provide any reference to support their claim). ***We believe this is also a contribution to control theory.***

4. **We added Reacher and Franka robot experiments and included it in Appendix G to address the reviewer's concern on adaptivity to Mujoco environments.**

---

---

> ### Author Response · Authors · 2025-12-03
> **Summary for Area Chair**
>
> ---
>
> ## 2. Reviewer x559 (rating 4)
> Reviewer x559 recognized the completeness of our literature review, our design for framework and especially, the ***efficiency of our proposed approach.***
>
> The most important concern Reviewer x559 raised is that they suggested our work overlaps with [2] (repeated 4 times in their comments) and we address this clearly in our response. **This reflects misunderstanding of what research questions each work addresses.** The reviewer also provided some more references for us to compare, and some of them fall outside of the same problem we address.
>
> **Our response:**
>  In rebuttal, we compared every aspect of our paper and [2], emphasized that these two papers are fundamentally different. We also compared each paper the reviewer proposed and clearly pointed out the significant difference.
>
> 1. ***[2] addresses existence: When is controllability preserved for bilinear Koopman models: under exact representation and sufficient data***. They provide theoretical ideal conditions only which are strong. The conditions cannot be easily verified or satisfied in any koopman framework. They provided **no learning algorithms, no neural networks, no results-relevant experiments**. On the other hand, ***We address design: How to learn controllable models from limited data for linear Koopman models?*** We provide canonical parameterization, regularizers, training algorithms, and demonstrated efficiency improvements from experiments.
> **Key distinction:** [2] proves controllability can exist under perfect conditions. We provide computational methods to enforce it during practical learning. These are fundamentally different contributions: knowing when something can exist doesn't provide tools to make it happen with real, limited data. These address fundamentally different research problems that would be recognized as distinct contributions. The claim "there exists some work that do it" by the reviewer is factually incorrect.
>
> 2. We compared every reference that the reviewer provided and show the fundamental difference with our work.
>
> 3. We added Reacher and Franka robot experiments and included it in Appendix G to address the reviewer's concern on adaptivity to more complex systems.
>
> We believe our response can clearly address the reviewer's concern.
>
> ---
>
> ## 3. Reviewer vF1q (rating 8)
> Reviewer vF1q recognized our contribution, stating: ***"From the perspective of deep learning-with-constraints literature, this paper provides a solid contribution that nicely integrates control-based results into design of deep learning architectures."*** ***as a response after our rebuttal.***
>
> **Our response:**
> 1. We added Reacher and Franka robot experiments and included it in Appendix G to address the reviewer's concern on adaptivity to more complex environments.
>
> 2. We provide detailed analysis showing modest 1.2–1.3× training overhead with the computation of Gramian. We use efficient coupled ODE implementations or Lyapunov equation that avoid explicit matrix exponentials integrals that the reviewer worried about. The computation can be even accelerated by using Van Loan's method. Consider that our approach itself has higher data efficiency, which means with much less data and much fewer training iterations, our methods can reduce the whole training resource in the end.
>
> ---

---

> ### Author Response · Authors · 2025-12-03
> **Summary for Area Chair**
>
> ---
>
> ## 4. Reviewer dLx9 (rating 6)
>
> Reviewer dLx9 correctly understood our contribution, recognizing that "embedding controllability into parameterization is novel" and that our "proofs linking OOC↔SOC...are mathematically well-grounded." This reviewer raised substantive technical questions about Koopman-specific concerns: expressive power trade-offs from canonical structure, computational cost of Gramian regularization, scalability to higher-dimensional systems, and connections to Koopman theory.
>
> **Our responses:**
>
> 1.  **Scalability to complex systems:** Added experiments on Reacher (MuJoCo) and 7-DoF Franka robot manipulator. Results show our controllability-based structure maintains effectiveness at higher complexity, with continued data efficiency advantages.
>
> 2. **Computational cost:** Detailed analysis showing modest 1.2–1.3× training overhead with the computation of Gramian. We use efficient coupled ODE implementations or Lyapunov equation that avoid explicit matrix exponentials integrals that the reviewer worried about. The computation can be even accelerated by using Van Loan's method. Consider that our approach itself has higher data efficiency, which means with much less data and much fewer training iterations, our methods can reduce the whole training resourse in the end.
>
> 3. **Expressive power trade-offs:** We discussed this issue in Appendix F of the paper. We already provided a relaxed way to design transformation matrix to improve expressivity. We show in our experiments, even with a more strict form, the proposed canonical parameterization produces Koopman models that exhibit higher fidelity than plain deep Koopman methods and MLPs.
>
> 4.  **Koopman theory connections:** We provide comprehensive discussion on Koopman-invariant subspaces with literature references showing that what we have done is not weaker than other approaches. Comparison with traditional EDMD using RBF dictionaries showing advantages of our neural approach. Also, compared with [3], we did not lose any constraints for equivariance.
>
> [1] Danhane, Baparou, Jérôme Lohéac, and Marc Jungers. "Characterizations of output controllability for LTI systems." Automatica 154 (2023): 111104.
>
> [2] Choi, Joonwon, Minhyun Cho, Hyunsang Park, Vishnu Vijay, and Inseok Hwang. "On The Controllability Preservation of Koopman Bilinear Surrogate Model." In 2024 IEEE 63rd Conference on Decision and Control (CDC), pp. 3457–3462. IEEE, 2024.
>
> [3] Cheng, Xiaoyuan, et al. "KEEC: Embed to Control on An Equivariant Geometry." arXiv preprint arXiv:2312.01544 (2023).
>
> ---
>
> # Summary
>
> Our contribution is clear: the first framework to enforce controllability by design in Koopman learning, with theoretical guarantees, reduced search space, and demonstrated superior data efficiency. We are tring make cross-disciplinary contribution: to control theory, deep learning with constraints, and Koopman learning.
>
> Revised manuscript uploaded with all improvements: new experiments (Appendix G), controllability clarifications (Introduction, Figure 4), computational analysis, expanded related work positioning.
>
> ---
>
> We hope this summary is helpful.
>
> Best regards,
>
> The Authors

---

### Meta-Review · Area_Chair_cb6Q · 2026-01-06

**Summary:**

This paper studies Koopman-based Neural ODE dynamics learning by incorporating "controllability" by design. Specifically, the paper proposes to enforce controllability by a canonical parameterization of the learned latent linear dynamics. An output Gramian regularizer was then added to further improve the degree of controllability. Overall, I found the approach clear and well-grounded, with strong empirical results. To my knowledge, the incorporation of Output-to-Output Controllability is novel. There were some concerns regarding the  novelty compared to existing works, the lack of experiments on more complex robotic tasks, which were mostly addressed by the rebuttal. I recommend that the authors incorporate the feedback from the reviewers in preparing the camera-ready version of the paper.

**Reviewer Concerns:**

The concerns regarding technical novelty, especially the comparison with State-to-State Controllability, the lack of more complex experiments, and the computational cost/overhead, are, in my opinion, adequately addressed. Other concerns regarding the novelty and "marginality" of the contributions, as well as more experiments on even more complex "robotic" tasks, are still relatively outstanding.

**Reviewer Scores:**

Reviewer ybHH, who gave 2, is more likely to increase their score due to the clarification on the technical novelty that explicitly compared OOC and SOC. Reviewer x559, who gave 4, is also likely to increase their score due to the addition of robotic experiments. Other reviewers who are already very positive will likely to maintain their positive ratings.

---

### Decision · Program_Chairs · 2026-01-26

Accept (Poster)